# RelationVLM: Making Large Vision-Language Models Understand Visual Relations

## Abstract

The development of Large Vision-Language Models (LVLMs) is striving to catch up with the success of Large Language Models (LLMs), yet it faces more challenges to be resolved. Very recent works enable LVLMs to localize object-level visual contents and ground text to them. Nonetheless, current LVLMs still struggle to precisely understand visual relations due to the lack of relevant data. In this work, we present RelationVLM, a large vision-language model capable of comprehending various levels and types of relations whether across multiple images or within a video. Specifically, we devise a multi-stage relation-aware training scheme and a series of corresponding data configuration strategies to bestow RelationVLM with the capabilities of understanding semantic relations, temporal associations and geometric transforms. Extensive case studies and quantitative evaluations show RelationVLM has strong capability in understanding such relations and emerges impressive in-context capability of reasoning from few-shot examples by comparison. This work fosters the advancements of LVLMs by enabling them to support a wider range of downstream applications toward artificial general intelligence.

## 1 Introduction

The success of Large Language Models (LLMs) (Brown et al., 2020; OpenAI, 2023; Touvron et al., 2023a) has advanced the development of large multimodal models, especially for Large Vision-Language Models (LVLMs) (Alayrac et al., 2022; Li et al., 2023b; Liu et al., 2023a). Recent studies (Zhang et al., 2023c; Wu et al., 2023b) enable the capabilities of understanding and processing the information from other modalities based on pre-trained LLMs, expanding their application ranges and opening countless possibilities in developing general-purpose visual understanding system.

Very recent research efforts on LVLMs (Liu et al., 2023a; Dai et al., 2023; Zhu et al., 2023; Peng et al., 2023) have been dedicated to enabling LVLMs to grasp correspondence between different modalities, exhibiting impressive capabilities on captioning and grounding. Nevertheless, compared to the enterprise-level performance of LLMs, the capabilities of understanding and processing multimodal signals by LVLMs are still in their early stages. Besides the correspondence between visual features and language, we also expect LVLMs to precisely understand the relations within the visual features themselves, thereby learning to make visual comparison as humans. This is, in fact, very critical for many downstream tasks in a wider range, *e.g.*, visual retrieval, anomaly detection, video understanding. By empirically evaluating the existing LVLMs as shown in Figure 1, including video-based LVLMs (Zhang et al., 2023b), we find they all struggle to precisely perceive and understand visual relations across different images or frames. Detailed comparison results are described in the caption of Figure 1.

Visual relations have a rich variety of types in nature, mainly including semantic relations (*i.e.*, whether visual objects have the same semantics), temporal associations (*i.e.*, the order of events in time), and geometric transforms (*i.e.*, spatial deformations). Building a LVLM capable of comprehensively understanding them (abbreviated as relation-aware LVLM ) is actually a challenging thing. This is because understanding these diverse visual relations requires handling data of higher dimensions than those handled by LLMs. Such capability is also hard to emerge directly after training with straightforwardly interleaved image-text pair data, as shown by the failure cases in Figure 1. Training such a LVLM from scratch requires an abundance of annotated image-text pairs, thus being

**Question:** What is the difference between these two pictures?

**RelationVLM (Ours):** The main difference between the two images is that there is a blue car $<img_1><loc_{440}><loc_{977}></img_1>$ in the second image, heading to the left and preparing to park in the parking lot, while it is not present in the first image. All other cars in the two images are the same.

**Minigpt4:** The first image shows a parking lot with several cars parked in it. The second image shows the same parking lot from a different angle, with the cars parked in different positions. The difference between the two images is that the second image shows the parking lot from a different perspective.

**Kosmos-2:** The image shows a parking lot with several cars parked in a row. There are at least ten cars visible in the image, with some closer to the left side of the lot and others towards the right side. The parking lot is surrounded by a fence, which is visible in both the top and bottom portions of the image. **(no difference)**

**LLaVA:** The difference between the two pictures is that one is a daytime view of a parking lot with cars parked in it, while the other is a nighttime view of the same parking lot with cars parked in it. The nighttime view shows the parking lot illuminated by streetlights, which provides a different perspective compared to the daytime view.

**InstructBLIP:** Cars are parked in parking lot in front of building. **(no difference)**

**Video-LLaMA:** There isn't a difference between the two pictures. They are a parking lot with cars. **(no difference)**

Figure 1: In a simple two-image comparison task, our RelationVLM highlights differences using grounded bounding boxes, while other methods miss correct differences (gray highlight) or claim no differences (marked with bold 'no difference'). '$<img_1>$' and '$</img_1>$' denote the second image, while '$<loc_{440}><loc_{977}>$' refers to the bounding box's left-top and right-bottom coordinates. More details see Sec. 3.2.

noteworthily costly. In this paper, we propose an efficient method for enabling LVLMs to understand visual relations with a pre-trained vision encoder and a pre-trained LLM as the language decoder. Moreover, we utilize the off-the-shelf annotations from the existing datasets with our proposed data configuration strategies, obviating the need for extra annotations about relations.

We introduce RelationVLM, a large vision-language model that not only has the grounding capability but also comprehends various visual relations. RelationVLM is built in the lowest costs as possible, by making full use of the knowledge already mastered by pre-trained models. Relation-VLM employs a pre-trained vision encoder, a pre-trained LLM and an adapter implemented by a linear layer, where we sorely need to train the adapter and fine-tune the LLM with LoRA (Hu et al., 2022) for reaching the desired goal. To achieve this, a three-stage training strategy is devised. In the first stage, we adopt coarsely interleaved image-text pairs for feature alignment across modalities. In the second stage, we extract relation-related attributes from the existing public datasets upon their available annotations, and automatically re-organize them into dialogue-form data using GPT-4 (OpenAI, 2023). RelationVLM grasp primary capabilities on understanding various relations through generative training with these data in this stage. Details on data creation and configuration are introduced in the method part. In the third stage, we perform instruction tuning with a combined dataset consisting of existing visual instruction tuning datasets and a high-quality subset manually selected from the dataset generated in the second stage.

After these three stages of training, our RelationVLM is not only able to accurately understand various relations across multiple images or within a video, but also demonstrates impressive in-context learning capabilities in unseen visual comparison application tasks, such as medical diagnosis and anomaly detection. By providing just a few examples in prompts, our model can robustly generalize its visual comparison capabilities and accurately apply them to specific domains even they are unseen before.

In summary, we make the following contributions in this work:

- We build RelationVLM, which addresses the shortcomings of current Large Vision-Language Models (LVLMs) in their inability to accurately comprehend various visual relations, including semantic relations, temporal associations and geometric transforms. It endows LVLMs with general-purpose visual comparison capabilities, being a step forward towards achieving general-purpose visual understanding system.

- We provide a data construction scheme for extracting relation attributes from existing public datasets and adopt a LLM (GPT-4) to automatically organize them into an appropriate form for multimodal generative training, for enabling LVLMs to comprehend various visual relations. And we introduce a cross-images relation instruction dataset for the first time.

- We qualitatively and quantitatively evaluate our built RelationVLM in comprehending different types of relations. Besides, we also showcase the visual in-context learning of our Relation-VLM for unseen visual comparison tasks, e.g., medical diagnosis and anomaly detection.

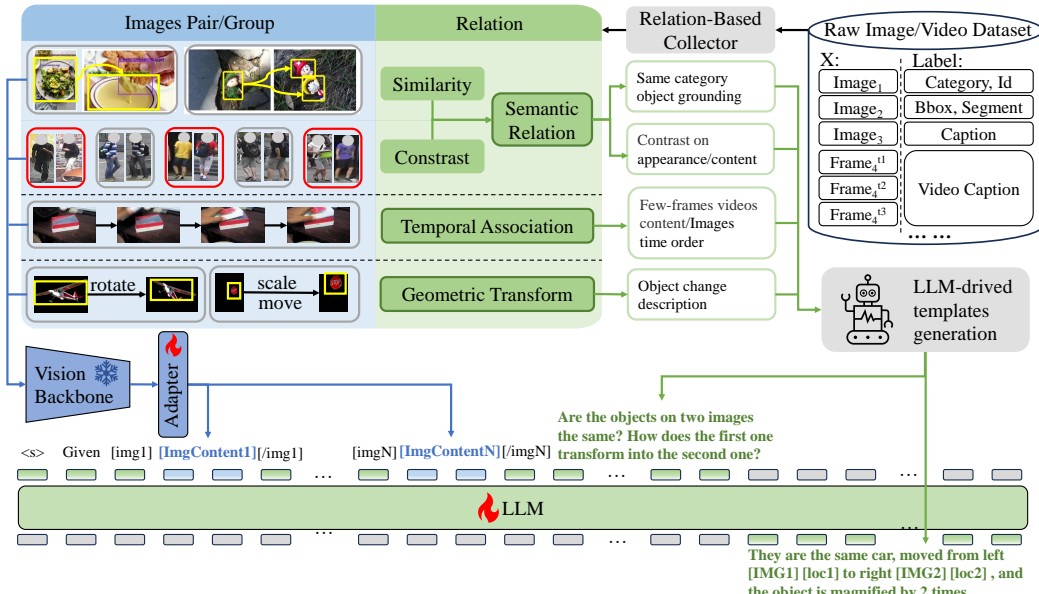

Figure 2: The overall training pipeline of RelationVLM.

## 2 RELATED WORK

### 2.1 LARGE VISION-LANGUAGE MODELS

**Image-based LVLMs.** Existing approaches on image-based LVLMs can be categorized into integrated vision-language interaction systems and end-to-end LVLMs. The former, such as Visual ChatGPT (Wu et al., 2023a), MM-REACT (Yang et al., 2023), HuggingGPT (Shen et al., 2023), and InternGPT (Liu et al., 2023c), integrate various existing vision models or tools into a centralized LLM controller with neural language prompts, without trainable parameters, excelling at well-defined problems but potentially lacking zero-shot abilities for open-ended instructions. The latter, including Flamingo (Alayrac et al., 2022), BLIP2 (Li et al., 2023b), MiniGPT-4 (Zhu et al., 2023), LLaVA (Zhang et al., 2023a), mPLUG-Owl (Ye et al., 2023), PaLM-E (Driess et al., 2023), KOSMOS-1/2 (Huang et al., 2023; Peng et al., 2023), employ vision encoders and finetune adapter layers for embedding alignment. And they either freeze LLM, finetune with LoRA (Hu et al., 2022) or end-to-end fine-tune LLM. Our work belongs to the latter, *i.e.* end-to-end LVLMs, but we introduce a novel training approach that significantly improves the model's capacity to comprehend relationships among multiple images, beyond other end-to-end LVLMs.

**Video-based LVLMs.** Video-based LVLMs focus on processing sequential video frames as input. Li et al. (2023c); Zhang et al. (2023b) integrates video foundational models with LLMs along with adapters. Maaz et al. (2023) introduces a novel annotation framework and proposes a quantitative video evaluating framework. Although, video-based LVLMs support multiple images as input, they exhibit certain limitations on understanding the visual relation. These models use frames derived from the same video, resulting in minimal scene variation and negligible changes between images. Consequently, relationships between images are predominantly limited to simple spatial motion. Additionally, the captions used for training often provide only coarse descriptions of temporal actions, lacking the fine-grained detail necessary for accurately depicting relationships between frames. In contrast, our work addresses not only temporal sequence relationships but also incorporates fine-grained semantic relations and geometric transforms which enables our model to capture more comprehensive and detailed relation information, surpassing the limitations of existing video-based LVLMs on understanding visual relations.

### 2.2 VISION-LANGUAGE IN-CONTEXT LEARNING

Vision-language in-context learning explores training-free few-shot learning, which often involves providing examples with input images and questions. Based on the method of providing examples, we can divide it into three categories: 1) **Language Prompts.** They provide examples as pure neural language, supplementing textual knowledge about new concepts, taking advantage of the in-context learning ability of LLMs. Huang et al. (2023); Peng et al. (2023) demonstrate good performance

in this area. However, only reliance on linguistic prompts may limit its flexibility and applicability. 2) **IQA Triplets Examples with Internal Association.** In this category, the provided examples are image-question-answer (IQA) triplets, which have dominated causal relationships within each triplet. Tsimpoukelli et al. (2021); Alayrac et al. (2022); Li et al. (2023a) enhance the model's capabilities by incorporating this type of examples during training. This method enables the model to learn causal relationships within IQA triplets, generalize to new images and questions, and generate answers, but does not capture input and example image relationships. 3) **IQA Triplets Examples with Cross-Triplet Association.** In this type, examples are IQA triplets with clear associations across example triplets and input triplet (input image), *e.g.* prototype classification, models need to compare input images with each prototype example image finely for a final response. This necessitates a robust capability comparison images' relation which Our RelationVLM have. In the experimental section, We demonstrate that our model achieves notable performance in this type of in-context learning task, outperforming previous approaches.

# 3 RELATIONVLM

## 3.1 DEFINITION OF VISUAL RELATIONS

We aim to build a LVLM that can understand visual relations. In general, visual relations can be divided into three main categories upon their intrinsic characteristics: semantic relations (similarity/contrast), temporal associations and geometric transforms. A certain type of visual relation may be manifested at different attributes. For example, the semantic relations of two objects may reside in their categories, colors, shapes, *etc.*. In this work, we extract relation annotations from existing public datasets including diverse semantic labels for categories, attributes, bounding boxes, *etc.*, and automatically organize them into a dialogue form for generative training.

To clearly describe our targeted relations and the construction process of their labels, we formulate a specific subdivided relation as a function denoted by $R_n$ where $n$ indexes the relation functions. Given a dataset $\mathcal{D} = \{(x_i, y_i)\}_{i=0}^{N}$ that includes $N$ images and $N_R$ types of relations in its annotations, we have a set of relation functions $\mathcal{R} = \{R_n\}_{n=0}^{N_R}$. Here, $x_i$ and $y_i$ represent a sample and its corresponding semantic label, respectively. $R_n$ maps the semantic labels of two given samples into a binary value, wherein $R_n(y_i, y_j) = 1$ denotes there exists the relation corresponding to $R_n$ between $x_i$ and $x_j$, and $R_n(y_i, y_j) = 0$ indicates there is no such relation. This definition is also applicable to image groups involving more than two images.

## 3.2 DATA CONSTRUCTION

**General introduction of the data construction process.** We introduce data construction for training RelationVLM, which is a dominant aspect for enabling LVLMs to understand various visual relations. In light of the enormous cost of annotating visual relations at scale, we extract the required labeled data from existing public datasets purposefully to cover different types of visual relations introduced in Sec.3.1. For semantic relations, we utilize the datasets containing reference expressions and spatial localization information for objects and persons, including GRIT (Peng et al., 2023), refCOCO (Yu et al., 2016), person reid datasets (Zheng et al., 2015; Li et al., 2014; Xiao et al., 2017; Wei et al., 2018; Li et al., 2015; Li et al., 2019). These datasets inherently carry explicit semantic annotations for entities, enabling the effortless acquisition of labels for semantic relationships between two entities based on the consistency of their labels, such as whether they belong to the same category or not. For temporal associations, we adopt the video datasets with captions for activities and the frame IDs that can describe the chronological orders. Specifically, we use SSv2 (Goyal et al., 2017) and WebVid (Bain et al., 2021). Regarding geometric transform, we cannot find off-the-shelf annotations for using. Thus, we segment natural images and perform geometric transformations (including Horizontal flip, vertical flip, brightness adjustment, rotation, scaling, and moving) on the segmented objects for synthesizing the needed dataset. Datasets for learning different types of visual relations are jointly used.

In Sec.3.1, given a dataset with off-the-shelf annotations, we formulate what image pairs or image groups can be considered to have a certain visual relation. To construct a new dataset containing diverse visual relations, we first collect the image pairs or groups with one or more defined visual relations from the aforementioned datasets. We encode the original annotations provided in the raw datasets into a linguistic form, and then utilize a mature LLM (GPT-4) to automatically gen-

erate natural language descriptions for visual relations via prompt engineering. Subsequently, we further adopt GPT-4 to further convert linguistic relation descriptions into a dialogue (*i.e.*, question-answering) form for generative training. Mathematically, taking an image pair $(x_i, x_j)$ as an example, such data construction process can be formulated as:

$$y_{i,j}^{dialog} = LLM^{dialog}(LLM^{desc}(E(y_i^{raw}), E(y_j^{raw}), p^{desc}|R_n), p^{dialog}|R_n), \tag{1}$$

where $y_i^{raw}$ and $y_j^{raw}$ are the raw annotations of $x_i$ and $x_j$, respectively. They satisfy the condition $R_n(y_i^{raw}, y_j^{raw}) = 1$, indicating that $x_i$ and $x_j$ have the relation corresponding to the relation function $R_n$. The $E(\cdot)$ represents the rule-based data processing function of encoding the raw annotations from their original form to be a linguistic form, $LLM^{desc}(\cdot)$ denotes the function of generating natural language captions with relation-related descriptions involved via a mature LLM, and $LLM^{dialog}(\cdot)$ denotes the function of converting natural language caption into a dialog form with a LLM. $p^{dialog}$ and $p^{desc}$ are the prompts for LLMs. They are manually designed and condition on the type of relation, *i.e.*, $R_n$. We detail them in the supplementary.

**Representative instantiation of data construction.** We generally introduce the data construction process as above. Here, we describe a representative example for further clarification. As shown in Figure 3, given two images $(x_i, x_j)$ in RAP (Li et al., 2015), each containing a unique individual, their raw annotations $(y_i^{raw}, y_j^{raw})$ provides the information of person IDs and a series of attributes. Let $R_{n1}(\cdot)$ be a relation function corresponding to the relation of different IDs in the sense that $R_{n1}(y_i^{raw}, y_j^{raw}) = 1$ indicates $x_i$ and $x_j$ are two different persons. Besides $R_{n1}(\cdot)$, we have other relation functions $R_n(\cdot)$ with different values of $n$ for describing other types of relations, such as the ones for clothing.

We first transform their raw annotations into a linguistic form individually using a rule-based data processing function, getting the results as follows:

> $E(y_i^{raw})$: person, ID: 128, male, short hair, short sleeve, short lower body clothing, pants, no hat, no backpack, no bag, no handbag, teenager, wearing a gray upper-body clothing, wearing black lower-body clothing.
> $E(y_j^{raw})$: person, ID: 334, male, short hair, short sleeve, short lower body clothing, pants, no hat, backpack, no bag, no handbag, teenager, wearing a black upper-body clothing, wearing black lower-body clothing, backpack bbox on $<loc_{256}> <loc_{489}>$.

The location tokens $<loc_{256}>$ and $<loc_{489}>$ are the locations of the top-left and bottom-right points of the bounding boxes, respectively. The encoding method for location tokens is from the commonly used one introduced in Peng et al. (2023); Chen et al. (2021), which represents the positions in an image by patchifying it into discretized bins.

Then, we prompt GPT-4 to automatically generate linguistic relation description for $(x_i, x_j)$ based on the relation function $R_n(\cdot)$ they satisfy. As formulated in Eq. 1, this process corresponds to $y_{i,j}^{desc} = LLM^{desc}(E(y_i^{raw}), E(y_j^{raw}), p^{desc}|R_n)$. The used prompts $p^{desc}|R_n$ are detailed in Sec C.4, and the obtained results are as below:

> $y_{i,j}^{desc}$: The main difference between them is the color of their upper-body clothing and the presence of a backpack. The first person is wearing a gray shirt while the second is wearing a black shirt. Additionally, the second person is carrying $<caption>$a backpack $</caption><img_1> <loc_{256}> <loc_{489}> </img_1>$, while the first person is not carrying any bags or backpacks.

In the illustrated $y_{i,j}^{desc}$ above, $<caption>$and $</caption>$are special tokens that indicate the beginning and the end of available semantic descriptions for the attachments. $<img_1>$ and $</img_1>$ tell which image the attachment belongs to. $<loc_{256}>$ and $<loc_{489}>$ indicate the location of the bounding box.

Furthermore, we convert $y_{i,j}^{desc}$ into a dialog (question-answering) form for generative training as Eq. 1. This step can enrich the produced data so as to improve the robustness and alignment of

RelationVLM. Detailed implementations of the used $p^{dialog}|R_n$ are provided in Sec. C.5. One of representative template for the finally generated $y_{i,j}^{dialog}$ is as below:

> $y_{i,j}^{dialog}$: ### Human: $<img_0> \{img\_content\}</img_0>$, $<img_1> \{img\_content\}</img_1>$ Are the two people in the two images the same person? $<grounding>$
> ### Assistant: No, they are not the same person. $\{y_{i,j}^{desc}\}$ Therefore, based on these differences, it can be concluded that these two people are not the same person.

Here, $<img_0>$ $</img_0>$ and $<img_1>$ $</img_1>$ points to different images. $\{img\_content\}$ is a placeholder, which will be replaced by the corresponding image tokens in the token space. The $<grounding>$ is a special token that informs RelationVLM that the subsequent answer should include grounding bounding boxes explicitly. As of now, one sample $(x_i, x_j), y_{i,j}^{dialog}$ for training RelationVLM is produced.

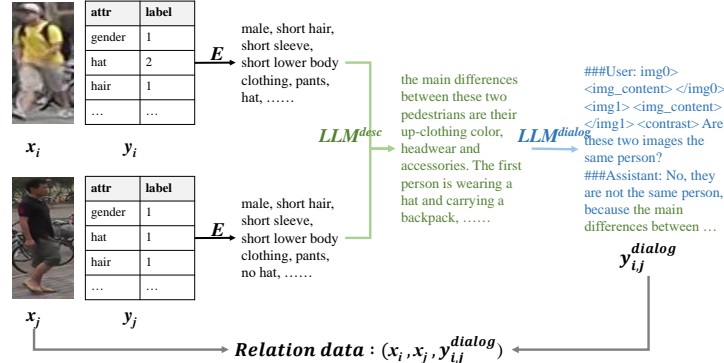

Figure 3: Illustration of an example of data construction.

## 3.3 MODEL TRAINING

As shown in Figure 2, a relation-based collector collects image pairs or groups from raw labeled datasets. Their original labels are input into LLM using prompts based on the relation type. The LLM generates fine-grained natural language descriptions along with referring bounding box. The image pairs/groups are then input into a frozen vision backbone to obtain vision tokens, which are transformed through an adapter layer and combined with language tokens for training the LLM using LoRA. RelationVLM consists of a vision encoder, an adapter and a LLM-based decoder, following the commonly used architecture design for LVLMs (Li et al., 2023b; Zhang et al., 2023a; Zhu et al., 2023; Ye et al., 2023). The vision encoder, composed of a VIT-based vision backbone and a Q-Former, encodes images into a set of visual tokens. The adapter achieves cross-modality alignment, which is implemented by a linear layer. The decoder could be a pre-trained LLM where we use Vicuna (Chiang et al., 2023) in this paper. To be efficient, we perform three-stage training for facilitating the learning wherein we make full use of the pre-trained weights for training RelationVLM as possible. The model training process can be delineated as below.

In Stage-1, we warm up the adapter with LAION-5B (Schuhmann et al., 2022), CC3M (Sharma et al., 2018), and SBU (Ordonez et al., 2011) for 50k iterations. We load the weights for the vision encoder from BLIP-2 (Li et al., 2023b) and load the weights of released Vicuna-7B for the LLM-based decoder in RelationVLM. In this stage, we freeze both the vision encoder and the LLM-based decoder. In Stage-2, the key stage for training RelationVLM, we continuously train the adapter and fine-tune the LLM-based decoder using LoRA (Hu et al., 2022; Mangrulkar et al., 2022) on the relation-contained data introduced in Sec.3.2, with the vision encoder frozen. In Stage-3, we perform instruction tuning for the adapter and the LLM-based decoder using LLaVA-instruct-150K dataset (Liu et al., 2023a), and MiniGPT4 IFT dataset (Zhu et al., 2023)), with the vision encoder still frozen. For all stages, we perform the typical generative training RelationVLM by maximizing the likelihood of each expected token based on preceding tokens over the decoding sequence. More details of training configurations are in Sec C.1.

## 4 EXPERIMENTS

### 4.1 EVALUATION METRIC

From a qualitative perspective, we explicitly showcase the capability of our proposed Relation-VLM in comprehending various visual relations. Besides, we also perform quantitative evaluation

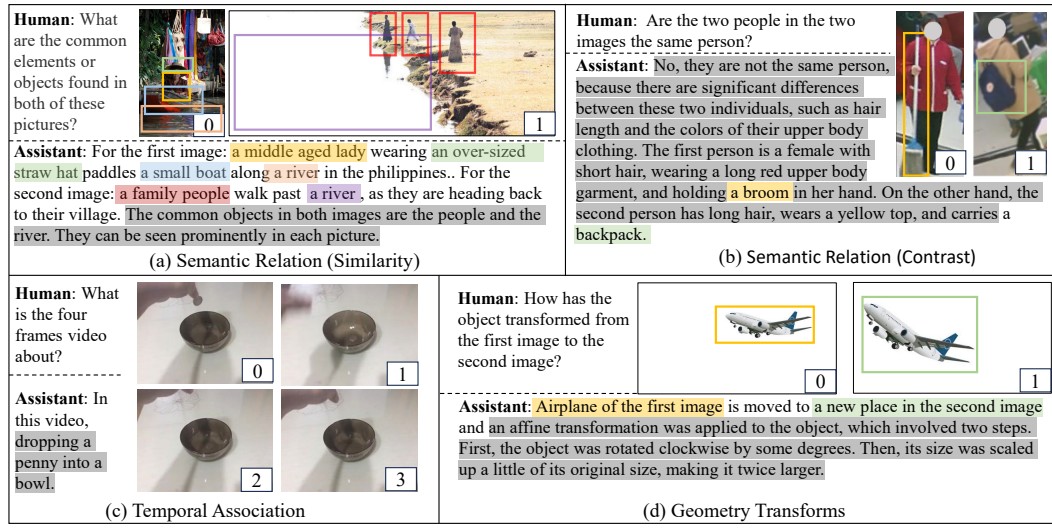

Figure 4: Examples of RelationVLM answering questions about image relations, including Semantic Relation (a)(b), Temporal Correlation (c), and Geometric Transformation (d). Color-highlighted text are output along with referring the same color bounding box. And the grey-highlighted text shows details relation description provided by RelationVLM.

with the following metrics: 1) **Traditional metrics.** We extract task-related from the outputs of RelationVLM, and use traditional metrics to assess their accuracies. For classification, we report the accuracy (Acc), precision (Prec), recall (Rec), and F1-score (F1). For grounding, following Yu et al. (2016); Liu et al. (2023b); Chen et al. (2023); Peng et al. (2023), we calculate the accuracy of predicted bounding boxes, named "BBox Acc". The predicted bounding box is considered correct when its IoU is greater than 0.5, and incorrect otherwise. It is considered as incorrect as well when it cannot be decoded successfully. 2) **LLM-based metric.** Considering that traditional metrics can only measure partial information of the model outputs, we employ a mature LLM (ChatGPT) to assess the overall plausibility of RelationVLM's outputs. We name this metric Relation Score (RS), and detail its use in the supplementary (Table 22). The range of RS is in [1-5] (the bigger RS, the better).

## 4.2 QUALITATIVE RESULTS

We show the qualitative results by providing representative case study results in Figure 4. As the example (a) shows, RelationVLM can not only perform grounding within a single image like Kosmos-2 (Peng et al., 2023), but also identify the contents of different images and accurately perceive the commonality between the two images. The example (b) presents that RelationVLM can distinguish between two different persons and further enumerate the differences in their attributes as the basis for judgment. The example (c) shows that RelationVLM is able to perceive the temporal association so as to recognize the happened activity in a video. The example (d) exhibits that RelationVLM is capable of accurately analyzing the geometric transformations that have occurred to the same object in the two images. These results explicitly demonstrate that our RelationVLM can understand different types of visual relations across multiple images or within a video.

## 4.3 QUANTITATIVE RESULTS

**Settings.** Considering the inference efficiency of LLMs, we sample some subsets from existing public datasets to evaluate the capabilities of RelationVLM in different aspects. Specifically, we use the subset sampled from the test set of COCO (Lin et al., 2014) to evaluate the capabilities of RelationVLM in identifying the common features among multiple images and referring expression comprehension (*i.e.*, grounding). Moreover, we construct a test set from the test sets of Xiao et al. (2017); Wei et al. (2018); Li et al. (2014); Zheng et al. (2015) to evaluate the capabilities

Table 1: Comparison with state-of-the-art (SOTA) grounding LVLMs on describing similarity. 'BBox Acc' represents the ability to locate the common objects, while 'RS' quantifies the quality of descriptions for similar objects.

| Method | BBox Acc (%) | RS |
|---|---|---|
| Shikra | 42.8 | 1.7 |
| Kosmos-2 | 41.3 | 1.6 |
| RelationVLM | **49.3** | **2.5** |

Table 2: Comparison with state-of-the-art (SOTA) person reid methods on describing contrast. 'Acc' represents the accuracy of determining if two images are of the same person, 'Prec' is an abbreviation for precision, 'Rec' refers to recall, and 'F1' denotes the F1 score. Meanwhile, 'RS' stands for 'Relation Score,' which quantifies the quality of detailed descriptions.

| Method | Acc(%) | Prec(%)) | Rec(%)) | F1(%) | RS |
|---|---|---|---|---|---|
| ResNet-50 (BOT) | 79.2 | 71.2 | **97.8** | 82.4 | - |
| ResNet-50 (SBS) | 89.8 | 85.4 | 96.0 | 90.4 | - |
| IBN-50 (MGN) | **92.2** | **91.4** | 93.2 | **92.3** | - |
| RelationVLM | 83.2 | 77.5 | 93.6 | 84.8 | **3.3** |

of RelationVLM in differentiating multiple similar images and identified their detailed differences. We construct a test set from SSv2 (Goyal et al., 2017) and ActivityNet (Fabian Caba Heilbron & Niebles, 2015) to assess the capability in understanding the temporal associations within videos, and construct another one from COCO (Lin et al., 2014) to evaluate the capability in understanding geometric transforms. More details about the datasets used for quantitative evaluation can be found in Supplementary (Table 21).

**Results.** In Table 1, we compare our RelationVLM with the State-Of-The-Art (SOTA) LVLMs *i.e.* Shikra (Chen et al., 2023) and Kosmos-2 (Peng et al., 2023) in their capabilities of grounding and identifying commonalities. Notably, the existing LVLMs Shikra and Kosmos-2 do not support input multiple images simultaneously within one inference process like ours. To enable comparison, we spatially concatenate multiple images into a single one as their input. As shown by the quantitative results, our RelationVLM can localize the targeted objects more accurately and identify commonalities better than other LVLMs. In Table 2, we compare our RelationVLM with SOTA task-specific expert models on person re-identification, including BoT (Luo et al., 2019), SBS (He et al., 2020), and MGN (Wang et al., 2018). Note that these models cannot provide linguistic answers like LVLMs but can provide distance measure results, we calculate the distances in the feature space using their released models[1] for 5,000 positive and negative sample pairs (with half of each), and use the average distance as the threshold. For the test sample pairs, we classify them as positive samples (*i.e.*, the same ID) if their feature space distance is less than the threshold, otherwise as negative samples (*i.e.*, different IDs). Based on the positive and negative sample classification results, we can obtain the numerical results for various metrics shown in Table 2. We observe that our RelationVLM outperforms the expert model BoT on accuracy and F1 score and achieves comparable results to the other two SBS and MGN. Note that these expert models are all trained with task-specific data and loss functions. Besides, they cannot provide natural language descriptions of the basis for judgment, *i.e.*, where the differences lie, as our RelationVLM. These results fully demonstrate the capability and superiority of our RelationVLM in visual comparison.

### 4.4 VISUAL IN-CONTEXT LEARNING PERFORMANCE

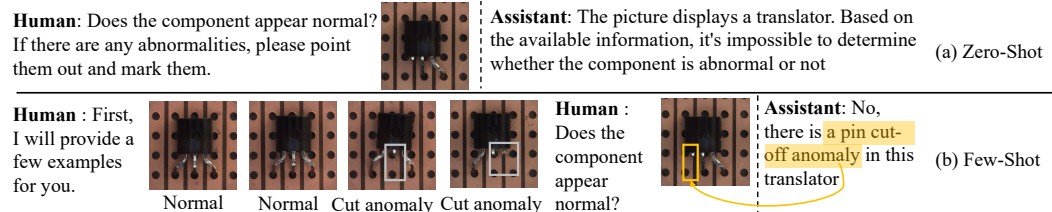

Figure 5: Illustration of RelationVLM's in-context learning performance on anomaly detection. (a) The zero-shot (no reference) result. (b) The few-shot (with few reference images provided) result.

---

[1]https://github.com/JDAI-CV/fast-reid/blob/master/MODEL_ZOO.md

**Settings.** We evaluate the visual in-context learning performance of RelationVLM on two unseen tasks: anomaly detection and medical diagnosis. We use the test data sampled from MVTec AD (Bergmann et al., 2019) and ISIC (Codella et al., 2018) for them respectively. We present the quantitative comparison results on these two tasks with the advanced LVLMs Flamingo (Alayrac et al., 2022) and Otter (Li et al., 2023a) in Table 3. Besides, we explicitly illustrate the representative case study results in Figure 5. More details on experiment configurations and more case study results are in Supplementary (Sec. B).

**Results.** As shown in Figure 5 (a), for anomaly detection, RelationVLM successfully recognizes the object within the given image. When no reference image is provided, our RelationVLM cannot determine whether the given electronic component in the displayed image is normal or anomalous, but can give a reasonable response to inform the user of what it knows. Once given a small number of reference images, as shown in Figure 5 (b), our model can precisely identify the specific type of anomaly, which fully demonstrates its excellent visual in-context learning performance on visual comparison. Impressive performance can also be observed on the task of medical diagnosis, with corresponding results placed in Supplementary (Sec. B). Furthermore, we compare our RelationVLM with two advanced LVLMs in their visual in-context learning performance. As shown in Table 3, when provided with different numbers of reference images, our RelationVLM consistently exhibits superior in-context learning performance compared to the other two LVLMs. Specifically, RelationVLM outperforms Otter, the second-best method, by 4.9% in accuracy for 8-shot settings on the MVTec AD dataset, and by 4.3% in the 8-shot setting on the ISIC dataset. These experimental results indicate that our RelationVLM has a stronger ability to understand diverse visual relations and is more adept at performing visual comparison.

Table 3: Quantitative results of comparing RelationVLM with SOTA LVLMs in their visual in-context learning performance. We report the accuracy (%) of 2/4/8-shot settings on MVTec AD and ISIC datasets for anomaly detection and medical diagnosis, respectively.

| Method | Acc(%)@Anomaly Detection | | | Acc(%)@Medical Diagnosis | | |
|---|---|---|---|---|---|---|
| | 2-shot | 4-shot | 8-shot | 2-shot | 4-shot | 8-shot |
| Open-Flamingo | 60.1 | 62.9 | 64.8 | 55.3 | 57.1 | 57.8 |
| Otter | 62.9 | 65.6 | 67.1 | 54.1 | 60.3 | 61.0 |
| RelationVLM | **66.8** | **71.1** | **72.0** | **55.5** | **63.6** | **65.3** |

## 5 CONCLUSION

In this paper, we present RelationVLM, a large vision-language model (LVLM) that addresses the limitations of current LVLMs in their inability of accurately understanding various visual relations, including semantic relations, temporal associations, and geometric transforms. RelationVLM expands the application scope of LVLMs, especially for tasks that require visual comparisons, taking an important step towards the realization of general-purpose visual understanding system. In more details, we introduce an efficient way to build RelationVLM, including a LLM-powered data construction scheme and a multi-stage model training strategy. The former extracts annotations related to diverse visual relations from existing public datasets and uses LLM to automatically convert them into a linguistic form for generative training. The latter aims to make full use of the knowledge already acquired by the pre-trained models and further learn how to understand various visual relations based on them, facilitating the learning. Furthermore, we comprehensively evaluate the enabled capabilities of RelationVLM. The qualitative case study results intuitively demonstrate that RelationVLM can accurately understand diverse visual relations and provide appealing natural language answers. Quantitative comparisons with expert models and existing advanced LVLMs further prove the superiority of our model in terms of visual comparison capabilities. Besides, we also evaluate the visual in-context learning performance of RelationVLM and observe favorable results on anomaly detection and medical diagnosis. We therefore believe that RelationVLM has great potential in advancing practical application of RelationVLM in the near future.

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

# A MORE EXPERIMENTAL RESULTS

## A.1 MORE QUANTITATIVE RESULTS

**Temporal and geometry.** In Table 4, we report the quantitative results on SSv2 and ActivityNet to assess the capability in understanding the temporal associations within videos, as well as the quantitative results on geometric test set to evaluate the capability in understanding the geometric transforms. We compare the SOTA video-based LVLMs , *i.e.* , Video-Chat (Li et al., 2023c), Video-LLaMA (Zhang et al., 2023b) and Video-ChatGPT (Maaz et al., 2023). For temporal association, as depicted in Table 4, RelationVLM achieves the best performance in both SSv2 test set and ActivityNet test set, particularly in video description tasks on the Something-Something v2 (SSv2) dataset, outperforming the second best, Video-ChatGPT by 1.3 RS. SSv2 demands that the model focuses more on temporal information and relationships, which underscores RelationVLM's superior ability to capture and comprehend temporal relations. For geometric transforms, as illustrated in Table 4, RelationVLM also achieves the best performance, which highlights the superior ability of RelationVLM to describe geometric transformations of objects. Furthermore, RelationVLM can track object movement with text grounding, a capability that the video LLMs we compared in Table 4 do not have. This combination of features fully demonstrates the superiority of our RelationVLM.

Table 4: Comparison with state-of-the-art (SOTA) video LVLMs on describing temporal association and geometric transforms.

| Method | Temporal Association | | Geometric Transforms | |
| | RS@SSv2 | RS@ActivityNet | Bbox Acc(%) | RS |
|---|---|---|---|---|
| Video-Chat | 1.7 | 2.2 | - | 1.2 |
| Video-LLaMA | 0.9 | 2.0 | - | 1.6 |
| Video-ChatGPT | 1.9 | 2.4 | - | 1.9 |
| RelationVLM | **3.2** | **2.4** | **69.7** | **3.2** |

**Overall comparison.** To ensure fairness, in Tables 1,2,3, and 4, we only compare models which either contain the same essential training datasets, such as person reid dataset and video dataset, or claim to have specific capacity for comparison, such as Grounding and visual in-context learning. In Table 5, we consolidate all metrics, including both relation metrics and in-context learning metrics, and all models, encompassing both state-of-the-art LVLMs and traditional task-specific experts, for a comprehensive comparison. Besides models in Table 1,2,3, and 4, we also include MiniGPT-4 (Zhu et al., 2023), LLaVA (Liu et al., 2023a), MM-GPT Gong et al. (2023), and mPLUG-Owl Ye et al.

(2023). As illustrated in Table5, our model, RelationVLM, surpasses all state-of-the-art LVLMs in all metrics, showcasing superior comprehensive relation understanding capability.

Table 5: Overall comprehensive comparison of various models across all metrics for relation understanding and in-context learning.

| Models | Relation Metric | | | | | | | | In-context | |
| | Similarity | | Contrast | | Temporal | | Geometry | | | |
| | Bbox Acc(%) | RS | Acc (%) | RS | RS@SSv2 | RS@AN. | Bbox Acc (%) | RS | 8-shot Acc@AD | 8-shot Acc@MD |
|---|---|---|---|---|---|---|---|---|---|---|
| BOT | - | - | 79.2 | - | - | - | - | - | - | - |
| SBS | - | - | 89.8 | - | - | - | - | - | - | - |
| MGN | - | - | **92.2** | - | - | - | - | - | - | - |
| MiniGPT-4 | - | 1.7 | 51.3 | 1.6 | 0.7 | 1.4 | - | 0.8 | 57.6 | 51.2 |
| LLaVA | - | 1.4 | 58.2 | 1.6 | 0.3 | 1.2 | - | 0.6 | 53.2 | 53.5 |
| MM-GPT | - | 1.2 | 53.2 | 1.1 | 0.3 | 1.1 | - | 0.5 | 54.6 | 52.4 |
| mPLUG-Owl | - | 1.2 | 56.2 | 1.5 | 0.9 | 1.6 | - | 0.5 | 53.9 | 54.1 |
| Shikra | 42.8 | 1.7 | 58.7 | 1.9 | 0.5 | 1.1 | 51.7 | 1.0 | 57.3 | 54.6 |
| Kosmos-2 | 41.3 | 1.6 | 60.2 | 1.8 | 0.5 | 1.2 | 55.3 | 1.1 | 59.7 | 55.8 |
| Video Chat | - | 0.7 | 50.9 | 1.3 | 1.7 | 2.2 | - | 1.2 | 50.3 | 51.1 |
| Video LLaMA | - | 0.9 | 52.1 | 1.5 | 0.9 | 2.0 | - | 1.6 | 52.3 | 51.7 |
| Video-ChatGPT | - | 1.1 | 54.3 | 1.6 | 1.9 | 2.4 | - | 1.9 | 52.5 | 52.4 |
| Open-Flamingo | - | 1.7 | 50.2 | 1.6 | 0.6 | 1.2 | - | 0.8 | 64.8 | 57.8 |
| Otter | - | 1.9 | 51.7 | 1.6 | 0.6 | 1.7 | - | 0.9 | 67.1 | 61.0 |
| RelationVLM | **49.3** | **2.5** | 83.2 | **3.3** | **3.2** | **2.4** | **69.7** | **3.2** | **72.0** | **65.3** |

**More evaluation on fine-grained images.** In Table 6, we present more quantitative results on CUB200-2011 (Wah et al., 2011) dataset to further evaluate the ability of our Relation-VLM in understanding contrast relations. We compare our RelationVLM with some traditional expert models of fine-grained classification task, including Part RCNN (Zhang et al., 2014), PS-CNN (Huang et al., 2016) and Bilinear-CNN (Lin et al., 2015). As these models cannot provide linguistic answers like LVLMs, we employ the same evaluation methods for these models as depicted as ReID models outlined in Table2.

Table 6: More contrast results on CUB200-2011 dataset. 'Acc' represents the accuracy of determining if two images are of the same kind of birds. 'RS' stands for 'Relation Score,' which quantifies the quality of detailed descriptions.

| Models | Acc (%) | RS |
|---|---|---|
| Part RCNN | 71.7 | - |
| PS-CNN | 73.4 | - |
| Bilinear-CNN | 81.3 | - |
| RelationVLM | **82.1** | **2.6** |

## A.2 ABLATION

We conduct an in-depth ablation study to investigate the impact of different training strategy on multiple images relation understanding. In this section, we follow the same evaluation method proposed in Sec. 4.2, we report the ablation studies in Table 7.

Table 7: The ablation study results, 'w.' stands for with, and 'w/o' denotes without. 'AN.' represents ActivityNet Dataset.

| Relation | Similarity | | Contrast | | Temporal | | Geometry | |
| | BBox Acc(%) | RS | Acc | RS | RS@SSv2 | RS@AN. | BBox Acc(%) | S. |
|---|---|---|---|---|---|---|---|---|
| w. LLaMA-2 | 49.3 | 2.5 | 78.3 | 3.2 | 3.1 | 2.1 | 61.9 | 2.7 |
| w/o LLM fine-tuning | 30.2 | 2.1 | 67.9 | 1.8 | 2.9 | 1.2 | 32.3 | 1.2 |
| w. Multi-tune Dialogue | 51.2 | 2.4 | 84.6 | 3.1 | 3.2 | 2.2 | 67.9 | 2.7 |
| Ours | 49.3 | 2.5 | 83.2 | 3.3 | 3.2 | 2.3 | 69.7 | 3.2 |

**Vicuna vs. LLaMA-2** We switch the LLM we used from Vicuna to LLaMA-2 (Touvron et al., 2023b) and observe a slight performance degradation, particularly in metrics related to traditional metric. This decrease might be due to the LLaMA-2 model not having undergone instruction tuning, while the Vicuna-based model follows the instruction better.

**Ablation of LLM tuning** Table 7 indicates that without LLM fine-tuning, performance significantly deteriorates, particularly in bounding box accuracy. Because the special tokens for grounding

are not encountered during LLM pretraining, which makes it challenging for the model to learn grounding without fine-tuning the LLM using LoRA. Furthermore, freezing the LLM reduces the number of trainable parameters, resulting in underfitting in complex relation comparison tasks.

**Ablation on Training data** We modify the dialog prompts on semantic contrast relation from a single-round dialogue format into a multi-rounds dialogue format by employing the "Two-Step Inquiry" approach, *i.e.*, asking the model twice: first for the judgment (whether it is right or wrong) and then for the details and reasons. This change increases the accuracy of the contrast, while slightly decreasing the relation score.

## A.3 SINGLE IMAGE QA EVALUATION

We also conduct evaluation on single image QA using POPE benchmark (Li et al., 2023d), and showcase our single image understanding on Sec. B. We compare state-of-the-art LVLMs trained to understanding single image content, including MiniGPT-4 (Zhu et al., 2023), LLaVA (Liu et al., 2023a), MM-GPT Gong et al. (2023), and mPLUG-Owl Ye et al. (2023), and find our Relation-VLM performs the best in this hallucination benchmark.

Table 8: Single image QA evaluation using POPE benchmark (Li et al., 2023d). "Yes Ratio" represents the probability of the model outputting a positive answer. Except for RelationVLM, the other results are obtained from Li et al. (2023d).

| Datasets | Metrics | RelationVLM(Ours) | MiniGPT-4 | LLaVA | MM-GPT | mPLUG-Owl |
|---|---|---|---|---|---|---|
| Random | Acc(%) (↑) | 84.97 | 79.67 | 50.37 | 50.10 | 53.97 |
| | Prec(%) (↑) | 89.40 | 78.24 | 50.19 | 50.05 | 52.07 |
| | Rec(%) (↑) | 79.27 | 82.20 | 99.13 | 100.00 | 99.60 |
| | F1(%) (↑) | 84.03 | 80.17 | 66.64 | 66.71 | 68.39 |
| | Yes Ratio (%) | 44.30 | 52.53 | 98.77 | 99.90 | 95.63 |
| Popular | Acc(%) (↑) | 81.43 | 69.73 | 49.87 | 50.00 | 50.90 |
| | Prec(%) (↑) | 83.55 | 65.86 | 49.93 | 50.00 | 50.46 |
| | Rec(%) (↑) | 78.20 | 81.93 | 99.27 | 100.00 | 99.40 |
| | F1(%) (↑) | 80.79 | 73.02 | 66.44 | 66.67 | 66.94 |
| | Yes Ratio (%) | 46.77 | 62.20 | 99.40 | 100.00 | 98.57 |
| Adversarial | Acc (%) (↑) | 79.77 | 65.17 | 49.70 | 50.00 | 50.67 |
| | Prec (%) (↑) | 77.60 | 61.19 | 49.85 | 50.00 | 50.34 |
| | Rec (%) (↑) | 83.60 | 82.93 | 99.07 | 100.00 | 99.33 |
| | F1 (%) (↑) | 80.49 | 70.42 | 66.32 | 66.67 | 66.82 |
| | Yes Ratio (%) | 53.83 | 67.77 | 99.37 | 100.00 | 98.67 |

## B CASE STUDIES

We illustrate a case study of medical diagnosis in Figure 6. More cases in which RelationVLM describes the relation are in Figure 7. And we also show a single image QA case, showing our model's ability to answer the question about a single image. In Figure 8, we show more cases of contrast relation description on fine-grained images.

## C IMPLEMENTATION DETAILS

### C.1 MORE CONFIGURATION DETAILS

### C.2 DATASET DETAILS

The training dataset details are shown in Table 11. As shown in Table 11, we use LAION-5B (Schuhmann et al., 2022), CC3M (Sharma et al., 2018), and SBU (Ordonez et al., 2011) on stage 1. We incorporated the GRIT dataset (Peng et al., 2023)), person reid dataset without attribute annotation (Zheng et al., 2015; Li et al., 2014; Xiao et al., 2017; Wei et al., 2018), person reid dataset with attribute annotation (Li et al., 2015; Li et al., 2019), Something-Something v2 (SSv2) dataset (Goyal

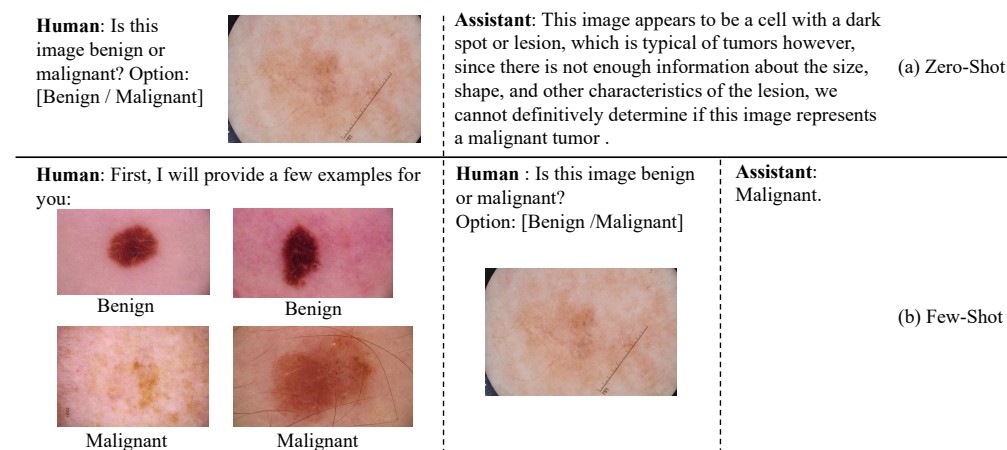

Figure 6: Illustration of RelationVLM's in-context learning performance on medical diagnosis. (a) The zero-shot (no reference) result. (b) The few-shot (with few reference images provided) result.

et al., 2017), WebVid dataset (Bain et al., 2021), and geometric transforms dataset. For stage 3 instruct-tuning, we used a subset of stage 2 data as well as the LLaVA-instruct-150K dataset (Liu et al., 2023a), and MiniGPT4 IFT dataset (Zhu et al., 2023)). We select the high-quality subset based on CLIP score, bbox confidence, and answer length. Specifically, we choose data where the CLIP score $> 0.34$ (if the CLIP score is available), bbox confidence $> 0.88$ (if bbox confidence is available), and text length $> 40$ words. Note that geometric transforms dataset is converted from COCO (Lin et al., 2014). We segment natural images and perform geometric transformations (including Horizontal flip, vertical flip, brightness adjustment, rotation, scaling, and moving) on the segmented objects for synthesizing the needed dataset, the corresponding label is the specific transformation applied.

### C.3 TRAINING DETAILS

We provide more detailed model configuration in Table 9 and detailed training configuration in Table 10. We train RelationVLM with Zero-2 powered by deepspeed framework (Rajbhandari et al., 2020; Rasley et al., 2020). The vision backbone has 1B parameters, and the vicuna model we used has 7B parameters, total number of trainable parameters is 0.5B. The model is trained using a $224 \times 224$ image resolution at all stages. We employ the AdamW Loshchilov & Hutter (2017) optimizer paired with a cosine learning rate scheduler Loshchilov & Hutter (2016) for model training. In the first stage, we conduct training over 100,000 steps using 4×A100 GPUs, with a global batch size of 96 and a base learning rate of 2e-3. This stage is completed in approximately 30 hours. In the second stage, the model is trained for 20,000 steps on 4×A100 GPUs, maintaining a global batch size of 96 and a base learning rate of 2e-4, taking around 6 hours. Finally, the model undergoes an additional 5,000 training steps on 4×A100 GPUs, with a global batch size of 64, completed in about 1.5 hours. The maximum learning rate remains constant at 2e-5 during this final stage.

### C.4 MORE DESCRIPTION PROMPTS ($p^{Desc}|R_n$)

Here we show more details about the description prompt ($p^{Desc}|R_n$ in Eq 1) in Table 12, Table 13 and Table 14.

### C.5 MORE DIALOG PROMPTS ($p^{dialog}|R_n$)

Here we show more details about the dialog prompt ($p^{Dialog}|R_n$ in Eq 1) in Table 15, Table 16, Table 18, Table 19,and Table 20.

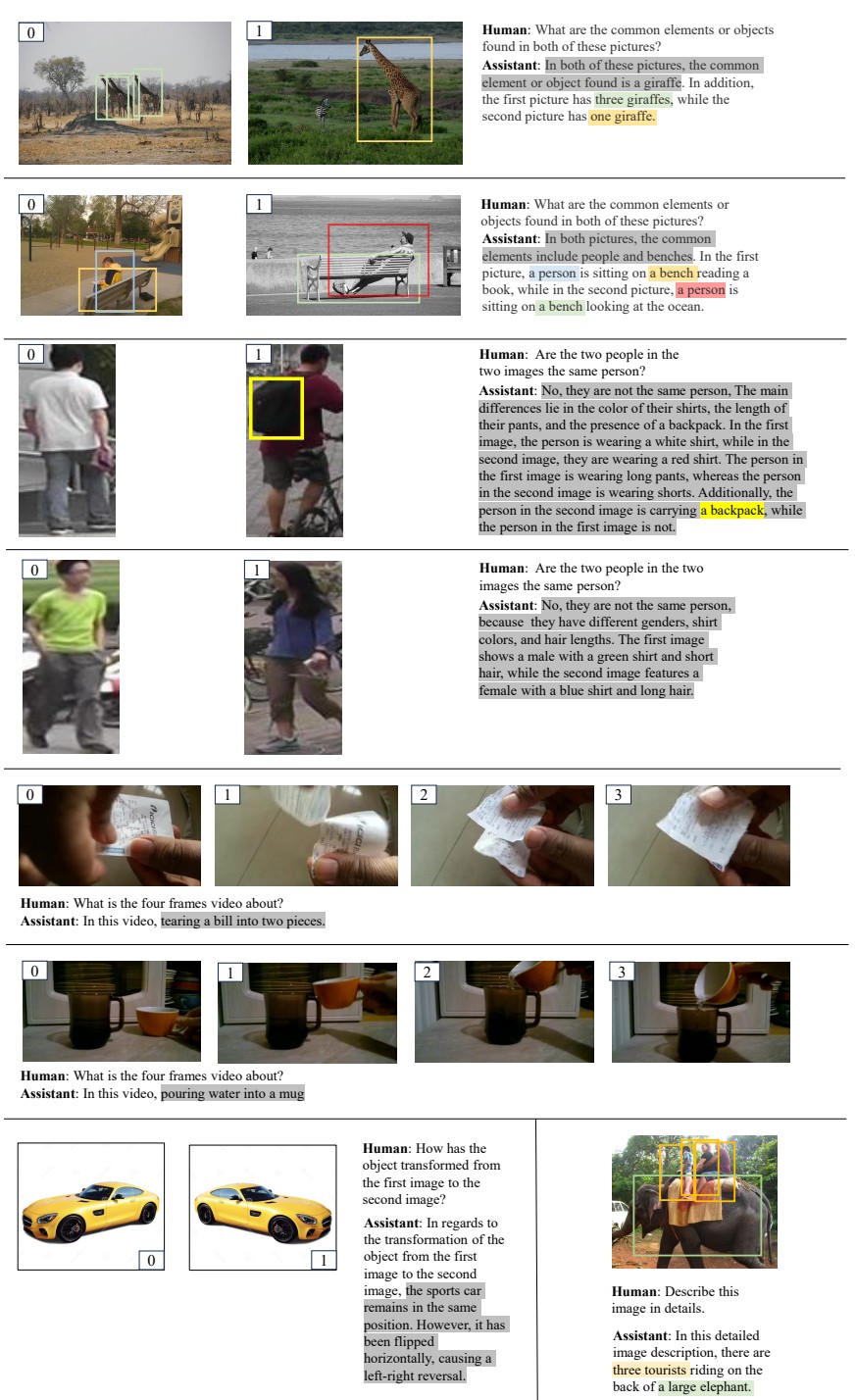

Figure 7: Illustration of more case studies. Color-highlighted text are output along with referring the same color bounding box. And the grey-highlighted text shows details relation description provided by RelationVLM. And we also showcase a single image QA case here.

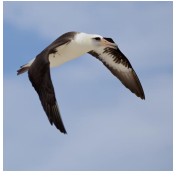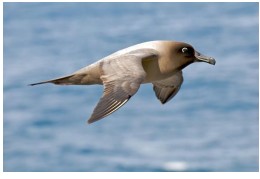

Human: What are the differences in their looks?

RelationVLM: The first bird features a white head, neck, and underparts with dark grey-brown upperparts, and a large, light pinkish-grey beak with a dark tip. The second bird has a uniform dark, sooty-brown plumage and a smaller dark grey bill with a bluish tinge. The first has a more contrasting color scheme, while colors of the other are more monochromatic.

Figure 8: Illustration of more contrast relation case studies on fine-grained images.

## C.6   EVALUATION DETAILS

We show details about the evaluation datasets on Table 21, and the relation score prompt in Table 22.

Table 9: Model configuration details on each stage. The snowflake icon represents the freezing of parameters in a particular stage, while the flame icon indicates fine-tuning of parameters during that stage.

| Model Configuration | Stage1 | Stage2 | Stage3 |
|---|---|---|---|
| Vision Backbone | ❄
BLIP-2 pretrain | ❄
BLIP-2 pretrain | ❄
BLIP-2 pretrain |
| Adapter | 🔥
From Scratch | 🔥
From Stage1 | 🔥
From Stage2 |
| LLM | ❄
Vicuna 7b | 🔥(LoRA)
Vicuna 7b | 🔥(LoRA)
From Stage2 |

Table 10: Training configuration details on each stage.

| Training Configuration | Stage1 | Stage2 | Stage3 |
|---|---|---|---|
| Base LR | 0.002 | 0.0002 | 0.00002 |
| Min. LR | 0.00008 | 0.00004 | 0.000008 |
| Warmup LR | 0.000001 | 0.000001 | 0.000001 |
| Warm-up scheduler | Linear | Linear | Linear |
| Scheduler | Cosine | Cosine | Cosine |
| Weight_decay | 0.01 | 0.05 | 0.05 |
| Training iterations | 50000 | 150000 | 10000 |
| Warmup iterations | 5000 | 10000 | 1000 |
| Lora_r | - | 8 | 16 |
| Lora_alpha | - | 32 | 32 |
| Lora_dropout | - | 0.1 | 0.1 |

Table 11: Training Datasets Details.

| Dataset | Labels | Relation Category | Description Prompt $(p^{Desc}|R_n)$ | Dialog Prompt $(p^{dialog}|R_n)$ |
|---|---|---|---|---|
| **Stage1** | | | | |
| LAION | Image Caption | - | - | Describe the image briefly. |
| CC3M | Image Caption | - | - | Describe the image briefly. |
| SBU | Image Caption | - | - | Describe the image. |
| **Stage2** | | | | |
| GRIT | Caption, Text-bbox Grounding | Semantic Similarity | Table 12 | Table 15 |
| refCOCO | Caption, Text-bbox Grounding | Semantic Similarity | Table 12 | Table 15 |
| PReID w/o Attr[1] | Person Ids | Semantic Contrast | - | Table 17 |
| PReID w. Attr[2] | Person Ids, Attribute Labels | Semantic Contrast | Table 13 | Table 16 |
| SSv2 | video Caption, Frame Ids | Temporal Association | - | Table 18, Table 19 |
| WebVid | video Caption, Frame Ids | Temporal Association | - | Table 18 |
| geometric transforms | Bbox, Transform Process | Geometry Transform | Table 14 | Table 20 |
| **Stage3** | | | | |
| Subset of Stage2 dataset[3] | All | All | All | All |
| LLaVA-instruct-150k | Dialog | - | - | [DIALOG][4] |
| MiniGPT4 IFT | Image Caption | - | - | Describe the image in detail. |

1. 'PReID w/o Attr' denotes person reid dataset without attribute annotation.
2. 'PReID w. Attr' denotes person reid dataset with attribute annotation.
3. Subset from stage 2 datasat, all dataset settings remain the same.
4. We split each dialog into rounds, and prompt the model to speak for the next round based on the dialog history.

Table 12: The description prompt ($p^{Desc}|R_n$) mention in Eq. 1 for GRIT and refCOCO dataset, $\{E(y_i^{raw})\}$ and $\{E(y_i^{raw})\}$ are the encoded linguistic annotations mention in Eq. 1

---

**System**:

I will provide the detailed captions of two images, along with each object's bounding box. Help me focus on the similarities between the images, as an explanation in natural language for why they are similar. Pay attention, you should remain the bounding box I give you ($e.g.$ $<img_0><loc_{24}><loc_{122}></img_0>$). First, I will give you an example:
Input:
Image 1: A beautiful beach scene with a palm tree, a boat, and an umbrella.
Bounding box 1: Palm tree $<img_0><loc_{24}><loc_{122}></img_0>$
Bounding box 2: Boat $<img_0><loc_{55}><loc_{326}></img_0>$
Bounding box 3: Umbrella $<img_0><loc_{97}><loc_{992}></img_0>$
Image 2: A relaxing beach view with a palm tree, a boat, and a sunbathing chair.
Bounding box 1: Palm tree $<img_1><loc_{66}><loc_{299}></img_1>$
Bounding box 2: Boat $<img_1><loc_{223}><loc_{597}></img_1>$
Bounding box 3: Sunbathing chair $<img_1><loc_{22}><loc_{736}></img_1>$
Output:
The two images share similarities, in the first image, there are $<caption>$a palm tree$</caption><img_0><loc_{24}><loc_{122}></img_0>$ and $<caption>$a boat$</caption><img_0><loc_{55}><loc_{326}></img_0>$ , as well as in the second image, there are also $<caption>$a palm tree$</caption><img_1><loc_{66}><loc_{299}></img_1>$ and $<caption>$a boat$</caption><img_1><loc_{223}><loc_{597}></img_1>$.
Now, here are the descriptions:

---

**User**:

Input:
Image 1: $\{E(y_i^{raw})\}$
Image 2: $\{E(y_j^{raw})\}$
Output:

---

Table 13: The description prompt ($p^{Desc}|R_n$) mention in Eq. 1 for person datasets with attribute annotation, $\{E(y_i^{raw})\}$ and $\{E(y_j^{raw})\}$ are the encoded linguistic annotations mention in Eq. 1

---

**System**:

I will provide the detailed descriptions of two pedestrians. Help me focus on the differences in their appearance, as an explanation in natural language for why they are not the same person. First, I will give you an example:
Input:
Person 1: male, short hair, short sleeve, short lower body clothing, pants, no hat, backpack, no bag, no handbag, teenager, wearing a red upper-body clothing, wearing black lower-body clothing
Person 2: female, short hair, short sleeve, short lower body clothing, pants, no hat, backpack, no bag, no handbag, teenager, wearing a white upper-body clothing, wearing black lower-body clothing
Output:
The main reasons we can tell they are not the same person are because of their gender and the color of their shirts. The first person is a boy wearing a red shirt, while the second person is a girl wearing a white shirt. These differences make it clear they are two separate individuals.
Now, here are the descriptions:

---

**User:**

Input:
Person 1: $\{E(y_i^{raw})\}$
Person 2: $\{E(y_j^{raw})\}$
Output:

---

Table 14: The description prompt ($p^{Desc}|R_n$) mention in Eq. 1 for geometric transforms dataset.

**System**:

I will provide the detailed descriptions of how object 1 has been transformed into object 2. Please describe the process in a simple and natural language. I will give you an example first:
Input:
Appled transformation: bounding box on Image1: $<loc_2>$ $<loc_{328}>$, bounding box on Image2: $<loc_{32}>$ $<loc_{358}>$, affine: a rotation of -20.74 degrees, a scale of 0.85.
Output:
$<caption>$The object $</caption><img_0><loc_2>$ $<loc_{328}></img_0>$ is moving to $<caption>$a new place $</caption><img_1><loc_{32}>$ $<loc_{358}></img_1>$. And an affine transformation was applied, which involved two steps. First, the object was rotated counterclockwise by 20.74 degrees. Then, its size was scaled down to 85% of its original size, making it slightly smaller. This combination of rotation and scaling changed the object's appearance while still maintaining its overall shape and proportions.
Now, here are the descriptions:

**User**:

Input:
Appled transformation: $\{E(y_{i,j}^{raw})\}$
Output:

Table 15: Examples of the dialog prompts ($p^{Dialog}|R_n$) mention in Eq. 1 for GRIT and refCOCO dataset. $\{y_{i,j}^{desc}\}$ is the relation description generated by GPT-4 in Eq 1. 'num_samples_needed' means the number of images needed for the current template.

---

### Human: $<img_0>$ {img_content}$</img_0>$ $<img_1>$ {img_content}$</img_1>$, What are the common elements or objects found in both of these pictures? $<grounding>$
### Assistant:Sure, $\{y_{i,j}^{desc}\}$
num_samples_needed: 2,

---

### Human: $<img_0>$ {img_content}$</img_0>$ $<img_1>$ {img_content}$</img_1>$, what do you see in common from these two images? $<grounding>$
### Assistant:$\{y_{i,j}^{desc}\}$
num_samples_needed: 2,

---

### Human: $<img_0>$ {img_content}$</img_0>$ $<img_1>$ {img_content}$</img_1>$, Can you identify the shared elements in these pictures? $<grounding>$
### Assistant:$\{y_{i,j}^{desc}\}$
num_samples_needed: 2,

---

### Human: $<img_0>$ {img_content}$</img_0>$ $<img_1>$ {img_content}$</img_1>$, What similarities can you find in these two images? $<grounding>$
### Assistant:$\{y_{i,j}^{desc}\}$
num_samples_needed: 2,

---

### Human: $<img_0>$ {img_content}$</img_0>$ $<img_1>$ {img_content}$</img_1>$ $<img_2>$ {img_content}$</img_2>$ Can you tell me the similarity of these images? $<grounding>$
### Assistant:Of course, $\{y_{i,j}^{desc}\}$
num_samples_needed: 3,

---

### Human: $<img_0>$ {img_content}$</img_0>$ $<img_1>$ {img_content}$</img_1>$ $<img_2>$ {img_content}$</img_2>$ What are the common features in these three images? $<grounding>$
### Assistant:$\{y_{i,j}^{desc}\}$
num_samples_needed: 3,

---

### Human: $<img_0>$ {img_content}$</img_0>$ $<img_1>$ {img_content}$</img_1>$ $<img_2>$ {img_content}$</img_2>$ Can you point out the similarities among these images? $<grounding>$
### Assistant:$\{y_{i,j}^{desc}\}$
num_samples_needed: 3,

---

### Human: $<img_0>$ {img_content}$</img_0>$ $<img_1>$ {img_content}$</img_1>$ $<img_2>$ {img_content}$</img_2>$ $<img_3>$ {img_content}$</img_3>$ What elements are present in all four of these pictures? $<grounding>$
### Assistant:$\{y_{i,j}^{desc}\}$
num_samples_needed: 4,

---

### Human: $<img_0>$ {img_content}$</img_0>$ $<img_1>$ {img_content}$</img_1>$ $<img_2>$ {img_content}$</img_2>$ $<img_3>$ {img_content}$</img_3>$ Can you find the common aspects in these four images? $<grounding>$
### Assistant:$\{y_{i,j}^{desc}\}$
num_samples_needed: 4,

---

Table 16: Examples of the dialog prompts $(p^{Dialog}|R_n)$ mention in Eq. 1 for person reid datasets with attribute annotations. $\{y_{i,j}^{desc}\}$ is the relation description generated by GPT-4 in Eq 1.'num_samples_needed' means the number of images needed for the current template. 'match' means the id matching results of two person images

---

### Human: $<img_0>$ {img_content}$</img_0>$, $<img_1>$ {img_content}$</img_1>$, Are these two images the same person, and why? $<grounding>$
### Assistant:Yes, they are the same person. $\{y_{i,j}^{desc}\}$
num_samples_needed: 2,
match: [0, 0],

---

### Human: $<img_0>$ {img_content}$</img_0>$, $<img_1>$ {img_content}$</img_1>$, Do these two images represent the same individual, and why? $<grounding>$
### Assistant:Yes, they represent the same individual. $\{y_{i,j}^{desc}\}$
num_samples_needed: 2,
match: [0, 0],

---

### Human: $<img_0>$ {img_content}$</img_0>$, $<img_1>$ {img_content}$</img_1>$, Can you tell me if these images are of the same individual, and why? $<grounding>$
### Assistant:Yes, they are the same individual. $\{y_{i,j}^{desc}\}$
num_samples_needed: 2,
match: [0, 0],

---

### Human: $<img_0>$ {img_content}$</img_0>$, $<img_1>$ {img_content}$</img_1>$, Are these two images the same person, and why? $<grounding>$,
### Assistant:No, they are the same person. $\{y_{i,j}^{desc}\}$
num_samples_needed: 2,
match: [0, 1],

---

### Human: $<img_0>$ {img_content}$</img_0>$, $<img_1>$ {img_content}$</img_1>$, Do these two images represent the same individual, and why? $<grounding>$,
### Assistant:No, they do not represent the same individual. $\{y_{i,j}^{desc}\}$
num_samples_needed: 2,
match: [0, 1],

---

### Human: $<img_0>$ {img_content}$</img_0>$, $<img_1>$ {img_content}$</img_1>$, Can you tell me if these images are of the same individual, and why? $<grounding>$,
### Assistant:No, they are not the same individual. $\{y_{i,j}^{desc}\}$
num_samples_needed: 2,
match: [0, 1],

Table 17: Examples of the dialog prompts $(p^{Dialog}|R_n)$ mention in Eq. 1 for person reid datasets without attribute annotations. 'num_samples_needed' means the number of images needed for the current template. 'match' means the id matching results of two person images

---

### Human: $<img_0>$ {img_content}$</img_0>$, $<img_1>$ {img_content}$</img_1>$, Are these two images the same person?
### Assistant:Yes, they are the same person.
num_samples_needed: 2,
match: [0, 0],

---

### Human: $<img_0>$ {img_content}$</img_0>$, $<img_1>$ {img_content}$</img_1>$, Do these two images represent the same individual?
### Assistant:Yes, they represent the same individual. ,
num_samples_needed: 2,
match: [0, 0],

---

### Human: $<img_0>$ {img_content}$</img_0>$, $<img_1>$ {img_content}$</img_1>$, Do these two images represent the same individual?
### Assistant:No, they do not represent the same individual. ,
num_samples_needed: 2,
match: [0, 1],

---

### Human: $<img_0>$ {img_content}$</img_0>$, $<img_1>$ {img_content}$</img_1>$, Can you tell me if these images are of the same individual?
### Assistant:No, they are not the same individual. ,
num_samples_needed: 2,
match: [0, 1],

---

### Human: $<img_0>$ {img_content}$</img_0>$, $<img_1>$ {img_content}$</img_1>$, $<img_2>$ {img_content}$</img_2>$, Are all of these images of the same person?
### Assistant:Yes, all of these images are of the same person. ,
num_samples_needed: 3,
match: [0, 0, 0],

---

### Human: $<img_0>$ {img_content}$</img_0>$, $<img_1>$ {img_content}$</img_1>$, $<img_2>$ {img_content}$</img_2>$, Are these three images the same person?
### Assistant:The first two is the same person, but the third one is not. ,
num_samples_needed: 3,
match: [0, 0, 1],

---

### Human: $<img_0>$ {img_content}$</img_0>$, $<img_1>$ {img_content}$</img_1>$, $<img_2>$ {img_content}$</img_2>$, Are all of these images of the same person?
### Assistant:No, none of these images are of the same person. ,
num_samples_needed: 3,
match: [0, 1, 2],

---

### Human: $<img_0>$ {img_content}$</img_0>$, $<img_1>$ {img_content}$</img_1>$, $<img_2>$ {img_content}$</img_2>$, $<img_3>$ {img_content}$</img_3>$, Are these four images of the same person?
### Assistant:The first, third, and fourth images are of the same person, but the second one is not.
num_samples_needed: 4,
match: [0, 1, 0, 0],

---

### Human: $<img_0>$ {img_content}$</img_0>$, $<img_1>$ {img_content}$</img_1>$, $<img_2>$ {img_content}$</img_2>$, $<img_3>$ {img_content}$</img_3>$, Are these four images of the same person?
### Assistant:The first and third images are of the same person, and the second and fourth images are of another person.
num_samples_needed: 4,
match: [0, 1, 0, 1],

Table 18: Examples of the dialog prompts ($p^{Dialog}|R_n$) mention in Eq. 1 for SSv2 dataset and WebVid dataset. $\{y^{raw}\}$ is the original video caption. 'num_samples_needed' means the number of frames needed for the current template.

---

### Human: $<img_0>$ {img_content}$</img_0>$, $<img_1>$ {img_content}$</img_1>$, $<img_2>$ {img_content}$</img_2>$, $<img_3>$ {img_content}$</img_3>$, what is the four frames video about?
### Assistant:In this video, $\{y^{raw}\}$,
num_samples_needed: 4,

---

### Human: $<img_0>$ {img_content}$</img_0>$, $<img_1>$ {img_content}$</img_1>$, $<img_2>$ {img_content}$</img_2>$, $<img_3>$ {img_content}$</img_3>$, can you describe these four images as if they were a video?
### Assistant:Sure, in this video, $\{y^{raw}\}$,
num_samples_needed: 4,

---

### Human: $<img_0>$ {img_content}$</img_0>$, $<img_1>$ {img_content}$</img_1>$, $<img_2>$ {img_content}$</img_2>$, $<img_3>$ {img_content}$</img_3>$, what does the four-images video show?
### Assistant:The video depict: $\{y^{raw}\}$,
num_samples_needed: 4,

---

Table 19: Examples of the dialog prompts ($p^{Dialog}|R_n$) mention in Eq. 1 for SSv2 dataset. $\{y^{raw}\}$ is the original video caption. 'num_samples_needed' means the number of frames needed for the current template.

---

### Human: $<img_0>$ {img_content}$</img_0>$, $<img_1>$ {img_content}$</img_1>$, $<img_2>$ {img_content}$</img_2>$, $<img_3>$ {img_content}$</img_3>$, the four frames video is about $\{y^{raw}\}$, what is the correct order of the input frames,
### Assistant:The corrent order is [2, 3, 0, 1],
order: [2, 3, 0, 1]
num_samples_needed: 4,

---

### Human: $<img_0>$ {img_content}$</img_0>$ $<img_1>$ {img_content}$</img_1>$ $<img_2>$ {img_content}$</img_2>$ $<img_3>$ {img_content}$</img_3>$, These four images are from a video with the original content: $\{y^{raw}\}$. Can you determine their correct sequence?
### Assistant:The correct order is [1, 3, 2, 0],
order: [1, 3, 2, 0]

---

### Human: $<img_0>$ {img_content}$</img_0>$ $<img_1>$ {img_content}$</img_1>$ $<img_2>$ {img_content}$</img_2>$ $<img_3>$ {img_content}$</img_3>$, these images are part of a video with the original content $\{y^{raw}\}$, can you put them in the right order?
### Assistant:The correct order is [2, 3, 0, 1],
order: [2, 3, 0, 1]

---

### Human: $<img_0>$ {img_content}$</img_0>$ $<img_1>$ {img_content}$</img_1>$ $<img_2>$ {img_content}$</img_2>$ $<img_3>$ {img_content}$</img_3>$, A video with the content $\{y^{raw}\}$ has these four frames. What is their proper arrangement?
### Assistant:The correct order is [3, 2, 0, 1],
order: [3, 2, 0, 1]

---

### Human: $<img_0>$ {img_content}$</img_0>$ $<img_1>$ {img_content}$</img_1>$ $<img_2>$ {img_content}$</img_2>$ $<img_3>$ {img_content}$</img_3>$, the video with content $\{y^{raw}\}$ consists of these four images. Can you arrange them correctly?
### Assistant:The correct order is [0, 2, 3, 1],
order: [0, 2, 3, 1]

---

Table 20: Examples of the dialog prompts ($p^{Dialog}|R_n$) mention in Eq. 1 for SSv2 dataset and WebVid dataset. $\{y^{raw}\}$ is the original video caption. 'num_samples_needed' means the number of frames needed for the current template.

---

### Human: $<img_0>$ {img_content}$</img_0>$, $<img_1>$ {img_content}$</img_1>$, How has the object transformed from the first image to the second image? $<grounding>$,
### Assistant:$\{y_{i,j}^{desc}\}$,
num_samples_needed: 2
### Human: $<img_0>$ {img_content}$</img_0>$ $<img_1>$ {img_content}$</img_1>$, What changes have occurred to the object between the first and second images? $<grounding>$
### Assistant:$\{y_{i,j}^{desc}\}$
num_samples_needed: 2

---

### Human: $<img_0>$ {img_content}$</img_0>$ $<img_1>$ {img_content}$</img_1>$, Can you describe the transformation that the object undergoes from the first image to the second image? $<grounding>$
### Assistant:$\{y_{i,j}^{desc}\}$
num_samples_needed: 2

---

### Human: $<img_0>$ {img_content}$</img_0>$ $<img_1>$ {img_content}$</img_1>$, What is the difference in the object's appearance between the first and second images? $<grounding>$
### Assistant:$\{y_{i,j}^{desc}\}$
num_samples_needed: 2

---

Table 21: Details of evaluation datasets and question. Using fixed questions stables the test results for better comparison (Li et al., 2023d).

| Benchmark | Dataset | Pairs Number | Evaluation Question |
|---|---|---|---|
| Similarity Relation | COCO | 1000 | What are the common elements or objects found in both of these pictures? |
| Contrast Relation | Market1501 CUHK03 cuhkSYSU MSMT17 | 5000 | Is the same person in these two images? And why? |
| Temporal Association | SSv2, ActivityNet | 5000 | What is the video about? |
| Geometry Transform | COCO | 1000 | How has the object transformed from the first image to the second image? |
| In Context | MVTec AD ISIC | 100 100 | Does the component appear normal? Is this image benign or malignant? option: [benign / malignant] |

Table 22: The relation score evaluation prompt.

**System**:

You are an intelligent chatbot designed for evaluating the factual accuracy of generative outputs for video-based question-answer pairs.
Your task is to compare the predicted answer with the correct answer and determine if they are factually consistent. Here's how you can accomplish the task:
##INSTRUCTIONS:
- Focus on the factual consistency between the predicted answer and the correct answer. The predicted answer should not contain any misinterpretations or misinformation.
- The predicted answer must be factually accurate and align with the content.
- Consider synonyms or paraphrases as valid matches.
- Evaluate the factual accuracy of the prediction compared to the answer.

**User**:

Please evaluate the following question-answer pair:
Question: $\{question\}$
Correct Answer: $\{y_{i,j}^{desc}\}$
Predicted Answer: $\{pred\}$
Provide your evaluation only as a factual accuracy score where the factual accuracy score is an integer value between 0 and 5, with 5 indicating the highest level of factual consistency.
Please generate the response in the form of a Python dictionary string with keys 'score', where its value is the factual accuracy score in INTEGER, not STRING.
DO NOT PROVIDE ANY OTHER OUTPUT TEXT OR EXPLANATION. Only provide the Python dictionary string.
For example, your response should look like this: "score': 4.8.

