# OpenReview forum: "RelationVLM: Making Large Vision-Language Models Understand Visual Relations"
_ICLR.cc/2024/Conference — Submitted to ICLR 2024_

### Official Review · Reviewer_qTNs · 2023-10-31

**Soundness:** 3 good
**Presentation:** 3 good
**Contribution:** 3 good
**Rating:** 6
**Confidence:** 5

**Summary:**

With the observation that existing LVLMs cannot find differences between pairs of images, the authors have RelationVLM that can understand visual relations. Specifically, they propose a new data construction scheme using an LLM to organize and generate dialogs. The authors evaluated their proposed RelationVLM both quantitatively and qualitatively. Finally, the authors demonstrated the performance in the few-shot and zero-shot settings.

**Strengths:**

1. The idea of automatically constructing dialogs from raw annotations to train RelationVLM is intriguing.

2. The manuscript is easy to read, and this reviewer enjoyed reading the paper.

3. The authors also demonstrated the performance of RelationVLM in the zero-shot and few-shot settings.

**Weaknesses:**

1. The authors introduced how to construct data for RelationVLM but did not explain the overall training. Figure 2 shows the overall training pipeline, but there are no sections or sentences that refer to Figure 2.

2. Relation Score is a new evaluation metric based on the assessment from an LLM. However, the evaluation cannot reply on an LLM as  RelationVLM was trained based on a dataset constructed based on an LLM-based decoder.

3. Minor issues:
* Figure 2: check the text color consistency in "Are the objects on two images the same?"
* Section 3.1: What is $N$ in $\mathcal{D} = \{(x_i , y_i )\}_{i=0}^N$ ?
* Section 3.2: `introduced in Sec.3.2`
* Page 5: $\mathcal{R}_{n1}(\cdot)$ any typo?
* quotation marks

**Questions:**

1. How do the authors handle multiple relations? Some pairs may contain more than one relation.

**Details Of Ethics Concerns:**

No ethical concerns. The authors constructed a dataset based on public datasets, and they hid the faces in figures.

---

> ### Author Response · Authors · 2023-11-22
> **Response to Reviewer qTNs**
>
> Thank you for your positive comments and valuable suggestions. We response to your concerns in detail as below.
>
> **Q1: The authors introduced how to construct data for RelationVLM but did not explain the overall training. Figure 2 shows the overall training pipeline, but there are no sections or sentences that refer to Figure 2.**
> **A1:** Thanks for this suggestion. We add overall training pipeline to Sec. 3.3 that refers to Figure 2 in our revision. Besides, the detailed descriptions for each training stage are also presented in this section.
>
> **Q2: Relation Score is a new evaluation metric based on the assessment from an LLM. However, the evaluation cannot reply on an LLM as RelationVLM was trained based on a dataset constructed based on an LLM-based decoder.**
> **A2:** In fact, evaluating the performance of large-scale models remains a highly challenging issue currently. We employ a LLM-based assessor for the model evaluation, following the current common practices [1][2][3] in this field. We will explore this aspect in our future study.
>
> **Q3: About the typos.**
> **A3:** Thanks for your reminder. We have fixed them in our revision.
>
> **Q4: What is the $N$ in $D = \{ (x_i, y_i) \}_{i=1}^N$?**
> **A4:** Sorry for this confusion. $N$ is the number of training samples. We add this description in the revision.
>
> **Q5: Does $R_{n1}(\cdot)$ on Page 5 have any typo?**
> **A5**: Sorry for this confusion. But we confirm that there is no typo in $R_{n1}(\cdot)$.
>
> **Q6: How do the authors handle multiple relations? Some pairs may contain more than one relations.**
> **A6**: In fact, a large number of such cases are already included in our current training data, i.e., some pairs contain one or more relations. In other words, one pair may have annotation for different relation types. It should also be noted that our model is an instruction-following agent. This allows us to handle a specified relation upon user instructions, even if several types exist at the same time.
>
> We look forward to your feedback on our responses. Please feel free to let us know if you have any new questions or suggestions.
>
> [1] Maaz, M., Rasheed, H., Khan, S. and Khan, F.S., 2023. Video-ChatGPT: Towards Detailed Video Understanding via Large Vision and Language Models. arXiv preprint arXiv:2306.05424.
> [2] Liu, H., Li, C., Wu, Q. and Lee, Y.J., 2023. Visual instruction tuning. arXiv preprint arXiv:2304.08485.
> [3] Li, X., Zhang, T., Dubois, Y., Taori, R., Gulrajani, I., Guestrin, C., Liang, P. and Hashimoto, T.B., 2023. Alpacaeval: An automatic evaluator of instruction-following models.

---

> > ### Comment · Reviewer_qTNs · 2023-11-23
> >
> > Thanks for the authors' responses. This reviewer will keep my original rating.

---

### Official Review · Reviewer_ZyhK · 2023-11-01

**Soundness:** 3 good
**Presentation:** 3 good
**Contribution:** 2 fair
**Rating:** 5
**Confidence:** 3

**Summary:**

The paper presents RelationVLM, a novel large vision-language model designed to understand a wide array of visual relations within images and videos. Addressing the limitations of existing Large Vision-Language Models (LVLMs), the authors propose a multi-stage relation-aware training scheme and data configuration strategies. RelationVLM excels in comprehending semantic relations, temporal associations, and geometric transforms, showcasing impressive in-context reasoning from few-shot examples. The model's capabilities are demonstrated through extensive evaluations, highlighting its proficiency in visual relation comprehension and in-context learning for novel visual comparison tasks. Key contributions include the development of RelationVLM, a unique data construction scheme for relation attribute extraction, and the advancement of LVLMs to support a broader range of applications, contributing to the progress toward artificial general intelligence.

**Strengths:**

1. The paper introduces RelationVLM, a novel large vision-language model specifically designed to comprehend a variety of visual relations across images and videos. This work addresses the limitations of existing Large Vision-Language Models (LVLMs) in understanding visual relations, proposing a multi-stage relation-aware training scheme and data configuration strategies as solutions. The model demonstrates strong capabilities in visual relation comprehension and impressive in-context reasoning from few-shot examples.

2. The research is backed by several evaluations and comparisons with existing LVLMs, showcasing the model's effectiveness and reliability. The authors provide detailed explanations of the model architecture, training procedures, and data construction scheme, ensuring reproducibility and transparency.

3. The paper is well-structured and written in a manner that makes it accessible to a wide audience, with clear explanations and examples provided to illustrate key concepts and methodologies.

4. The contributions of this paper are significant, as it advances the capabilities of LVLMs in understanding visual relations, supporting a broader range of applications, and moving closer to achieving artificial general intelligence. The development of RelationVLM, along with the unique data construction scheme for relation attribute extraction, represents a substantial step forward in the field.

**Weaknesses:**

1. The paper could enhance its validation of RelationVLM by extending the range of benchmarks and comparisons with existing models, particularly those that are considered state-of-the-art in the field of vision-language models. This would provide a more solid foundation for assessing the model's performance and capabilities.

2. The process of data construction is central to the training of RelationVLM, yet the paper does not delve into potential biases that might be introduced during this phase. A thorough analysis of data diversity and strategies to mitigate bias would contribute to the robustness and reliability of the model.

3. The complexity of the model architecture and training scheme necessitates a discussion on the computational resources required, as well as the scalability and efficiency of the model across different settings and applications.

4. The paper aims to enhance the model's comprehension of visual relations, but the definitions and explanations of these relations are somewhat concise. Expanding on the types of visual relations, along with providing more examples, would offer clearer insights into the model's understanding and categorization of these relations.

5. The evaluations presented primarily focus on controlled settings. Incorporating assessments of the model's performance in real-world scenarios would demonstrate its applicability and effectiveness outside of experimental conditions.

6. The paper would benefit from a more comprehensive discussion on the limitations of the proposed model and approach, as well as potential areas for future research and development. This would provide a balanced perspective and guide subsequent efforts in advancing the field.

**Questions:**

1. Could you provide more information on the choice of benchmarks for evaluating RelationVLM? Including additional benchmarks, especially those involving state-of-the-art models, could strengthen the validation of RelationVLM's capabilities.

2. How does the data construction process account for potential biases, and what steps were taken to ensure data diversity? A detailed explanation would enhance the robustness of the model and ensure the generalizability of the results.

3. Can you elaborate on the computational resources required for RelationVLM, and discuss its scalability and efficiency across different settings?

4. The paper briefly explains different types of visual relations. Could you provide a more detailed taxonomy and additional examples to offer clearer insights into how the model comprehends and categorizes these relations?

5. Are there evaluations of RelationVLM in real-world scenarios or applications?

6. Could you provide a more thorough discussion on the limitations of RelationVLM and the proposed approach, as well as potential areas for future work?

7. How does RelationVLM handle ambiguous or unclear visual relations in images or videos?

---

> ### Author Response · Authors · 2023-11-22
> **Response to Reviewer ZyhK (1/2)**
>
> Thank you for praising the novelty, effectiveness, reliability, writing quality of our work and the significance of our contributions. For your concerns, we response to them in detail as follows.
>
> **Q1: Extending the range of benchmarks and comparisons with existing models.**
> **A1:** Thanks for this suggestion. We have performed a broader comparison over extended benchmarks with more models. The corresponding results are **in Table 5 of our revision**.
>
> **Q2: A thorough analysis of data diversity and strategies to mitigate bias would contribute to the robustness and reliability of the model.**
> **A2:** Thank you for this valuable suggestion. Currently, the diversity and bias of our data depend to some extent on GPT-4, which is recognized as the most advanced large-scale model. In fact, evaluating the performance of large-scale models, including the diversity and bias of their data, remains a highly challenging issue. We have not yet found a scientific evaluation method from existing publications, and will explore this issue in our future research.
>
> **Q3: The complexity of the model architecture and training scheme necessitates a discussion on the computational resources required, as well as the scalability and efficiency of the model across different settings and applications.**
> **A3:** We detail the complexity analysis as follows. The vision backbone has 1B parameters, and the vicuna model we used has 7B parameters, totally number of trainable parameters is 0.5B.  In the first stage, we conduct training over 100,000 steps using 4$\times$A100 GPUs, with a global batch size of 96 and a base learning rate of 2e-3. This stage is completed in approximately 30 hours. In the second stage, the model is trained for 20,000 steps on 4$\times$A100 GPUs, maintaining a global batch size of 96 and a base learning rate of 2e-4, taking around 6 hours. Finally, the model undergoes an additional 5,000 training steps on 4$\times$A100 GPUs, with a global batch size of 64, completed in about 1.5 hour. The maximum learning rate remains constant at 2e-5 during this final stage. We have added this analysis in our revised supplementary materials.
>
> **Q4: The paper aims to enhance the model's comprehension of visual relations, but the definitions and explanations of these relations are somewhat concise. Expanding on the types of visual relations, along with providing more examples, would offer clearer insights into the model's understanding and categorization of these relations.**
> **A4:** Thanks for this valuable suggestion as well. Our work is the first one of its kind to explore how to enable the capabilities of understanding visual relations for LVLMs. As an initial study, we start with the three most basic and widely applied high-level categories of visual relations. Here, we further clarify their definitions. For the first class, semantic relation, there are the similarity and the contrast. Among them, the similarity refers to the degree for a given object of being similar to the other one in another image in terms of their category. If they are of the same category, the model should ground them spatially. The contrast is defined as the difference between two instances in terms of their category as well as a series of semantic attributes. The temporal association and the geometric transform refer to the motions or transforms over time and in spatial, respectively. We will attempt to expand the types of visual relations in our future investigation.
>
> **Q5: The evaluations presented primarily focus on controlled settings. Incorporating assessments of the model's performance in real-world scenarios would demonstrate its applicability and effectiveness outside of experimental conditions.**
> **A5:** Our in-context learning evaluation on anomaly detection or medical diagnosis could be viewed as in-the-wild assessment in real-world scenarios, as their corresponding data and tasks are unseen for the model training. Besides, we will release a model demo for more assessments in the wild once completing our internal reviews.

---

> ### Author Response · Authors · 2023-11-22
> **Response to Reviewer ZyhK (2/2)**
>
> **Q6: The paper would benefit from a more comprehensive discussion on the limitations of the proposed model and approach, as well as potential areas for future research and development. This would provide a balanced perspective and guide subsequent efforts in advancing the field.**
> **A6:** Thanks for this suggestion. One of our current main limitations lies in the data used for handling geometric transform is synthesized by us, rather than being collected from the real world. We have not yet found any publicly annotated data that can be utilized, so we synthesize them. We will further improve our model in this aspect in the future study.
>
> **Q7: How does RelationVLM handle ambiguous or unclear visual relations in images or videos?**
> **A7:** As an initial exploration for teaching LVLMs to understand visual relations, we make our covered relation as basic and widely used as possible to ensure the generalization ability in different scenarios. For ambiguous or unclear visual relations that beyond our modelling range, frankly speaking, we are currently unable to make sure how the model performs. This may require case-by-case discussions, and is worthy of a further exploration in the future.
>
> We look forward to your feedback on our responses. Please feel free to let us know if you have any new questions or suggestions.

---

> > ### Comment · Reviewer_ZyhK · 2023-11-22
> >
> > Thank you for addressing my comments and queries. Having reviewed your responses and considering the insights from other reviews, I've chosen to maintain my initial score.

---

### Official Review · Reviewer_izCF · 2023-11-04

**Soundness:** 3 good
**Presentation:** 3 good
**Contribution:** 2 fair
**Rating:** 3
**Confidence:** 4

**Summary:**

The paper aims to improve relation understanding capabilities of large vision-language models (VLM). The types of relations they consider includes semantic relations, temporal associations and geometric transformations. They propose to curate instruction tuning data to improve these kinds of relations by feeding ground-truth data into the GPT model. With the help of curated instruction tuning data, the proposed RelationVLM outperforms competing methods on several benchmarks.

**Strengths:**

- The paper points out the issues of existing VLMs' relation understanding, namely semantic relations, temporal associations and geometric transformations.
- The detailed description about data curation approach using GPT is valuable.

**Weaknesses:**

- After works like "When and why vision-language models behave like bags-of-words, and what to do about it?", it is well-known that VLMs are weak in relation detection. Afterwards, there have been a few works in this domain. It is absolutely crucial to compare the proposed methods against (simple extension of) existing methods.
- It is unclear if the improved performance is due to more data or the curated relation-aware instruction tuning data. It would be great if you could demonstrate that the existing models' performance does not improve by adding more data. That way, you can prove that we need special training data. Also, for each VLM, it would be helpful if the datasets use for RelationVLM, e.g. SSv2, ActivityNet are used or not.

Some minor points:
- What is the baseline approach in Table 1? Is it Shikra or Kosmos-2?
- Why do you think anomaly detection is a good benchmark for showcasing the benefit from this approach? Please explain the link between anomaly detection and relation understanding.

**Questions:**

Since the method and problem the paper is addressing are kind of well-known, I would expect thorough experimental analysis from this paper. Please address points listed in "Weaknesses" section for the next version. If backed up by more experiments, I think this work could be accepted to other top-tier conferences.

---

> ### Author Response · Authors · 2023-11-21
> **Response to Reviewer izCF**
>
> Thank you for your postive comments and valuable suggestions. We provide detailed responses for addressing your concerns below.
>
> **Q1: Comparison with CLIP-based VLMs like "When and why vision-language models behave like bags-of-words, and what to do about it?"**
> **A1:** Thank you for this suggestion. The work you list belongs to CLIP-based VLMs. We would like to point out that it's actually diffcult to compare ours with them on most evaluation tasks we used since they lack the capability to handle language input prompts to execute multiple different relation tasks, respond with language output to describe relations, or perform grounding to locate relations. Despite this, we still make a preliminary comparison with them in a comparable experimental setting. Specifically, we evaluate these CLIP-based VLMs in our contrast task using a method similar to the one we used to perform the comparisons with traditional reid models in our experiments. The comparison results are reported as follows:
> | Models   | CLIP[1] | NegCLIP[2] | VFC[3] | RelationVLM(Ours) |
> |----------|:-------:|:----------:|:------:|:-----------------:|
> | Acc (\%) | 60.5    | 65.4       | 66.3   | **83.2**          |
>
> **Q2: It is unclear if the improved performance is due to more data or the curated relation-aware instruction tuning data.**
> **A2:** The improved performance mainly comes from our curated relation-aware instruction tuning data, instead of more data. We draw this conclusion based on our experiments in Table 1, 2, 4, wherein all model are trained on the same training data sources while their differences lie in that our model has adopted our proposed relation-aware data curation strategies but others do not.
> In other words, the performance improvements benefit from the data curation strategy instead of the data itself.
>
> **Q3: What is the baseline approach in Table 1? Is it Shikra or Kosmos-2?**
> **A3:** The results in Table 1 correspond to comparisons at the model level, instead of an ablation study. Both Shika and Kosmos-2 should not be viewed as our baseline models in the ablation study.
>
> **Q4: Please explain the link between anomaly detection and relation understanding.**
> **A4:** We apply the task of anomaly detection for evaluating the capability of in-context learning with a few-shot evaluation setting. In this setting, as shown in Figure 5, normal and anomaly samples are given as references. When performing anomaly detection for a given sample, a visual comparison is necessitated between the test sample and the reference samples, in which visual relations are required to be captured for completing such comparison.
>
> We look forward to your feedback on our responses. Please feel free to let us know if you have any new questions or suggestions.
>
> [1] Radford, A., Kim, J.W., Hallacy, C., Ramesh, A., Goh, G., Agarwal, S., Sastry, G., Askell, A., Mishkin, P., Clark, J. and Krueger, G., 2021, July. Learning transferable visual models from natural language supervision. In International conference on machine learning (pp. 8748-8763). PMLR.
>
> [2] Yuksekgonul, M., Bianchi, F., Kalluri, P., Jurafsky, D. and Zou, J., 2022, September. When and Why Vision-Language Models Behave like Bags-Of-Words, and What to Do About It?. In The Eleventh International Conference on Learning Representations.
>
> [3] Momeni, L., Caron, M., Nagrani, A., Zisserman, A. and Schmid, C., 2023. Verbs in action: Improving verb understanding in video-language models. In Proceedings of the IEEE/CVF International Conference on Computer Vision (pp. 15579-15591).

---

### Official Review · Reviewer_BRZ9 · 2023-11-07

**Soundness:** 2 fair
**Presentation:** 2 fair
**Contribution:** 3 good
**Rating:** 5
**Confidence:** 4

**Summary:**

This paper studies the visual relationship in vision-language tasks using LVLM. Specifically, the authors constructed a large-scale visual relationship dataset to train the LVLM. By combining the visual features of two images encoded by a vision model, LVLM is leveraged to output description to capture the visual relationship between two images. In the output, LVLM is trained to answer specifically on the detailed changes and would point out the difference location. By comparing the proposed method to other existing LVLMs, the authors prove that their model is largely enhanced to analyze visual relationships.

**Strengths:**

- This paper studies an interesting problem, it is a novel contribution and could have great potential usage.
- The qualitative performance is great. Based on the presented examples, each image is carefully described, and the relationship is correctly demonstrated.

**Weaknesses:**

- This paper is not well-written, which could be further improved.
- I suggest some claims should have proper literature, experimental, or theoretical support. The claim that “Nonetheless, current LVLMs still struggle to precisely understand visual relations” has no clear evidence in the abstract, which is odd for me during reading. Moreover, it is not rigorous to assume only three factors affect the visual relations: “characteristics: semantic relations (similarity/contrast), temporal associations and geometric transforms.” It is highly possible that other factors such as corruption, lighting conditions, etc could have an impact on the perception difference between two images.
- The baseline comparison is not enough. There are still many other strong LVLMs are not considered, such as LLaVA, MiniGPT-4, mPLUG-OWL, etc. Besides, why some tables have different comparisons? Some take KOSMOS as a baseline, others take Openflamingo as a baseline, which is quite confusing to me and is not a fair comparison.

**Questions:**

Please see weaknesses.

---

> ### Author Response · Authors · 2023-11-21
> **Response to Reviewer BRZ9**
>
> Thank you for your positive comments and valuable suggestions. We response to your concerns in detail as below.
>
> **Q1: About the paper writing.**
> **A1:** Thanks for pointing this out. We have revised our paper by fixing several typos, simplifying the superscript and subscript of equations, adding additional experimental results, and re-organizing the experiments section in the supplementary materials. Please check the specific changes in the revision.
>
> **Q2: About the evidences for the statement "Nonetheless, current LVLMs still struggle to precisely understand visual relations".**
> **A2:** For this statement, we propose intuitive illustration in Figure 1, as well as numerical evidences in Table 1, Table 2 and Table 4 in our paper. We demonstrate this through the shown inferiority of them when compared to our proposed model in understanding different types of visual relations.
>
> **Q3: It is not rigorous to assume only three factors affect the visual relations.**
> **A3:** In fact, ***we do not make such an assumption, nor do we study the factors affecting the visual relations in this paper.*** The semantic relations, temporal associations and geometric transforms are the ***three main types / categories*** of relations that we define based on the characteristics of visual relations.
>
> **Q4: Why are some LVLMs still not considered (such as LLaVA, MiniGPT-4, mPLUG-OWL, etc.)?**
> **A4:** In this work, we aim to enable the LVLMs with new capabilities in understanding visual relations that these model does not have. Thus, when we evaluate these target new capabilities, these LVLMs cannot deliver comparable results with ours for specific comparison. We add their corresponding evaluation results **in Table 5 of our revision**. These experimental results indicate that these models are very deficient in these capabilities. Besides these results, please also not that we have considered the comparisons with these models on the POPE benchmark un Table 8 of our supplementary since they are comparable on this benchmark.
>
> **Q5: Why do some tables have different models for comparisons?**
> **A5:** We evaluate different capabilities by performing the corresponding comparisons in different tables. To ensure the fairness of the comparison, we only compare our model with the ones that use the same training data in these aspects.
>
> We look forward to your feedback on our responses. Please feel free to let us know if you have any new questions or suggestions.

---

> > ### Comment · Reviewer_BRZ9 · 2023-11-22
> > **Thanks for your reply**
> >
> > Thanks for the comments, I have carefully read the revised paper. After some consideration, I decided to keep my score.

---

### Official Review · Reviewer_zaJj · 2023-11-07

**Soundness:** 3 good
**Presentation:** 2 fair
**Contribution:** 2 fair
**Rating:** 5
**Confidence:** 3

**Summary:**

This paper studied the visual relation understanding across images/frames based on the recent LLMs and VLMs. By formulating the cross-image/frame visual relation understanding into a dialog problem, this work re-organized the existing datasets, adopted the existing off-the-shell models to build a new one for the above task, and achieved improvements on several tasks and benchmarks.

**Strengths:**

+ The visual relation understanding matters for many downstream tasks, and the setting of this work is sound.

+ The dataset re-organization and curation may be useful for the community.

+ Overall, the whole paper and method are easy to follow.

+ The training designs and metrics are reasonable based on existing works.

**Weaknesses:**

- Lacking many essential details to understand the proposed method and data set fully:

Any bias analysis of the text generated from GPT?

The high-quality subset was manually picked: any details of the cost, quality, and process?

How to ensure the rationality of the geometric transformation?

I saw the examples, it seems that the geometric transformation cases are with a blank background.

- Better illustration:

Eq. 1: the superscript and subscript are all too complex.

The data curation process needs a visualized process.

The best results in the tables can be bold.

- Though there were several tables of results, are their scale and generalization enough to support the claim?

- There are many controversies. But I think we still need to be careful about using the word AGI, especially without the discussion of the path to its precise description/definition and the relation between this work and AGI.

**Questions:**

1. Though it is just a case, in Fig. 1, the shadow also differs.

2. Possible testing on visual relation understanding within one image/frame? Like two objects/persons in an image.

3. Tab 6: the Rec and Yes Ratio show disadvantages, any discussion?

---

> ### Author Response · Authors · 2023-11-21
> **Response to Reviewer zaJj (1/2)**
>
> Thank you for your positive comments and valuable suggestions. We response to your concerns in detail below.
>
> **Q1: Any bias analysis of the text generated from GPT?**
> **A1:** We apologize that we have no idea on how to conduct a bias analysis, and we also have not found relevant analysis in related publications for reference. If possible, we hope you could share specific requirements and methods for bias analysis to help us in this regard. Thank you!
>
> **Q2: The high-quality subset was manually picked: any details of the cost, quality, and process?**
> **A2:** We select the high-quality subset based on the CLIP score, bbox confidence, and answer length. Specifically, we opt for data where the CLIP score is greater than 0.34 (if the CLIP score is available), the bounding box confidence is greater than 0.88 (if the bounding box confidence is available), and the text length exceeds 40 words. We have provided these details in the revision.
>
> **Q3: How to ensure the rationality of the geometric transformation?**
> **A3:** As an initial exploration, we have not found off-the-shelf annotated datasets to use regarding geometric transform. Therefore, we segment natural images and apply random basic geometric transformations to the segmented objects for synthesizing the corresponding data. Although the dataset is synthesized, we strive to maintain a broad diversity of the synthesized data to enhance the generalization capability of the model as much as possible.
>
> **Q4: Better illustration.**
> **A4:** Thank you for your valuable suggestions. In the revision, we simplify the superscript and subscript in Eq.1 and add  Figure 3 for visualizing the data construction. Besides, we also highlight the best results in experimental results in bold as you suggest.
>
> **Q5: On scaling and generalization performance of experimental results.**
> **A5:** Given that the evaluation of large-scale models is still a problem worth studying, here, we attempt to conduct some empirical experiments to further analyze the scaling and generalization performance of our experiment results. In terms of the scaling performance, we scale up the test set for evaluating the capability of grounding the objects of the same category to five times its original size. Its corresponding test results are reported in the table below. We can observe that our proposed RelationVLM consistently outperforms other LVLMs on this scaled test set. Regarding the generalization performance, we provide the few-shot results evaluated in the wild on real-world scenarios such as anomaly detection and medical diagnosis, in Figures 5, 6, and Table 3.
> | Method | BBox Acc (\%) | RS           |
> |-----------------------------|:-------------:|:------------:|
> | MiniGPT-4                   | -             | 1.3          |
> | LLaVA                       | -             | 1.5          |
> | MM-GPT                      | -             | 1.6          |
> | mPLUG-Owl                   | -             | 1.7          |
> | Shikra                      | 43.3          | 1.8          |
> | Kosmos-2                    | 42.4          | 1.7          |
> | RelationVLM(Ours)  | **49.6** | **2.6** |
>
> **Q6: Be careful about using the word AGI.**
> **A6:** Thank you for the reminder. We have refined relavant statements by removing this word in the revision. We aim to narrow the gap between the capabilities of existing models and the anticipated AGI ambitions by compensating for the capabilities of existing LVLMs.
>
> **Q7: In Fig. 1, the shadow also differs.**
> **A7:** Yes, this is a failure case of ours. As the first attempt to explore how to enhance LVLMs' understanding of visual relevance, our model has not yet reached the human level. And there are still some failures on challenging samples that have subtle differences.
>
> **Q8: About the performance of visual relation understanding within one image/frame.**
> **A8:** Our proposed model can support the visual relation understanding within one image/frame. To demonstrate this, we conduct a validation experiment by combining two images of pedestrians into one and input it into the model for testing the accuracy of judging whether the two people on the left and right are the same person. The results are as follows:
> | Models   | MiniGPT-4 | LLaVA | RelationVLM(Ours) |
> |----------|:---------:|:-----:|:-----------------:|
> | Acc (\%) | 59.3      | 57.4  | **80.2** |

---

> ### Author Response · Authors · 2023-11-21
> **Response to Reviewer zaJj (2/2)**
>
> **Q9: In Tab 6, the Yes Ratio and Rec. show disadvantages, any discussion?**
> **A9:** It should be noted that the Yes Ratio is not necessarily better when it's higher or lower. Thus, the results do not indicate disadvantages. We follow [1] to measure the bias degree of our proposed model with this metric on its public benchmark in which the answers Yes and No account for half each. So a Yes Ratio near 50\% indicates a less biased model. Here, our RelationVLM outperforms other LVLMs in this regard. Besides, a lower Rec. (Recall) do not indicate the disadvantages as well. Balancing recall with precision is crucial to the effectiveness of any system. Overemphasizing one aspect can compromise the other. For instance, our model has a lower recall compared to LLaVA, which is a result of LLaVA's near 100\% Yes Ratio. This implies LLaVA's inclination to answer 'Yes' to all questions, which may inflate its recall, but doesn't necessarily indicate enhanced performance, considering its lower accuracy, precision, and F1 score. On the other hand, our model excels in both accuracy and F1 scores, demonstrating its overall superior performance. Therefore, despite the lower recall, our model's balanced performance metrics signify its effectiveness.
>
> We look forward to your feedback on our responses. Please feel free to let us know if you have any new questions or suggestions.
>
> [1] Li, Yifan, et al. "Evaluating object hallucination in large vision-language models." In EMNLP, 2023.

---

> > ### Comment · Reviewer_zaJj · 2023-11-22
> > **Post-rebuttal**
> >
> > Thanks for the response. After reading the reviews and responses, I tend to keep my rating.

---

### Official Review · Reviewer_hRT4 · 2023-11-07

**Soundness:** 3 good
**Presentation:** 3 good
**Contribution:** 3 good
**Rating:** 6
**Confidence:** 3

**Summary:**

This paper aims to adopt LVLMs to learn visual relations among images. This paper first constructed a dataset for image relation learning based on conventional vision or vision-language datasets. RelationVLM, a vision encoder followed by an adapter and an LLM-based decoder, has been proposed to learn image relations based on the constructed dataset.

**Strengths:**

- This paper has explored an interesting task,  cross-image visual relations comparison, and constructed a large-scale dataset for this task. The dataset construction pipeline is interesting.

- The proposed method has achieved a more comprehensive relation analysis compared to conventional LVLMs.

- This paper is well-organized and easy to follow.

**Weaknesses:**

- The architecture of the proposed method, RelationVLM, is trivial. The core contribution of this paper may be a dataset contribution scheme. However, in the title and abstract, I cannot see a description of this core contribution.

- The experiments may be not sufficient. If the visual relations comparison could benefit other visual tasks, e.g., image retrieval or fine-grain classification, such experiments should be conducted to further prove the meaning and value of the fine-grain relation or difference comparison of two images.

**Questions:**

See weaknesses.

---

> ### Author Response · Authors · 2023-11-21
> **Response to Reviewer hRT4**
>
> We appreciate your positive comments on our targeted problem, proposed method and paper writing. We response to your suggestions in detail below.
>
> **Q1: Regarding highlighting the contributions on data construction.**
> **A1:** Thanks for your suggestion in this regard. We indeed compensate for the deficiency of current large vision-language models in understanding visual relations, based on commonly used model architectures. We have highlight this in the revision, marked in red.
>
>
> **Q2: Could the visual relation comparison benefit other visual tasks, e.g., image retrieval or fine-grained classification?**
> **A2:** Thank you for this question. We conduct a fine-grained classification experiment on CUB200-2011 dataset, main results of which are presented in the following table. We add these results in Table 6 and illustrate a representative case study in Figure 8 of our revised version. These results demonstrate that our proposed RelationVLM outperforms SOTA LVLMs and some traditional task-specific models by a clear margin.
>
> | Models   | Part RCNN | PS-CNN | Bilinear-CNN | MiniGPT-4 | LLaVA | RelationVLM (Ours)          |
> |----------|:---------:|:------:|:------------:|:---------:|:-----:|:--------:|
> | Acc (\%) | 76.4      | 76.6   | 85.1         | 20.3      | 19.4  | **85.7** |
>
> We look forward to your feedback on our responses. Please feel free to let us know if you have any new questions or suggestions.

---

> > ### Comment · Reviewer_hRT4 · 2023-11-23
> > **Post-rebuttal**
> >
> > Thanks for the authors' rebuttals. After reading the authors' responses and other reviewers' comments, I tend to keep my original rating.

---

### Meta-Review · Area_Chair_doKA · 2023-12-11

**Metareview:**

This paper was reviewed by three experts and received mixed scores. Though all reviewers agree some aspects of the paper are promising, they also consistently raise concerns listed below.

1. The technical contribution of the proposed model is incremental (hRT4).

2. The clarity of presentations needs to be improved (hRT4, zaJj).

 3. The experiments are limited. More ablation studies and comparisons are required to verify the model (hRT4, BRZ9, ZyhK).

While the research demonstrated indeed has promise, the decision is not to recommend acceptance in its current state. The authors are encouraged to consider the reviewers' comments when revising the paper for submission elsewhere.

**Justification For Why Not Higher Score:**

1. The technical contribution of the proposed model is incremental (hRT4,).

2. The clarity of presentations needs to be improved (hRT4, zaJj).

 3. The experiments are limited. More ablation studies and comparisons are required to verify the model (hRT4, BRZ9, ZyhK).

**Justification For Why Not Lower Score:**

NA

---

### Decision · Program_Chairs · 2024-01-16

Reject